# Geodynamic and seismotectonic model of a long-lived transverse structure: The Schio-Vicenza Fault System (NE Italy)

Dario Zampieri[1], Paola Vannoli[2], Pierfrancesco Burrato[2]

[1]Dipartimento di Geoscienze, Università degli Studi di Padova, Padova, Italy
[2]Istituto Nazionale di Geofisica e Vulcanologia, Sezione Roma 1, Rome, Italy

*Correspondence to*: Pierfrancesco Burrato (pierfrancesco.burrato@ingv.it)

**Abstract.** We make a thorough review of geological and seismological data on the long-lived Schio-Vicenza Fault System (SVFS) in northern Italy and present for it a geodynamic and seismotectonic interpretation.

The SVFS is a major and high angle structure transverse to the mean trend of the Eastern Southern Alps fold-and-thrust belt, and the knowledge of this structure is deeply rooted in the geological literature and spans for more than a century and a half. The main fault of the SVFS is the Schio-Vicenza Fault (SVF), which has a significant imprint in the landscape across the Eastern Southern Alps and the Veneto-Friuli foreland. The SVF can be divided into a northern segment, extending into the chain north of Schio and mapped up to the Adige Valley, and a southern one, coinciding with the SVF proper. The latter segment borders to the east the Lessini, Berici Mountains and Euganei Hills block, separating this foreland structural high from the Veneto-Friuli foreland, and continues southeastward beneath the recent sediments of the plain via the blind Conselve-Pomposa fault. The structures forming the SVFS have been active with different tectonic phases and different style of faulting at least since the Mesozoic, with a long-term dip-slip component of faulting well defined and, on the contrary, the horizontal component of the movement not well constrained. The SVFS interrupts the continuity of the Eastern Southern Alps thrust fronts in the Veneto sector, suggesting that it played a passive role in controlling the geometry of the active thrust belt and possibly the current distribution of seismic release. As a whole, apart from moderate seismicity along the northern segment and few geological observations along the southern one, there is little evidence to constrain the recent activity of the SVFS. In this context, the SVFS, and specifically its SVF strand, has accommodated a different amount of shortening of adjacent domains of the Adriatic (Dolomites) indenter by internal deformation produced by lateral variation in strength, related to Permian - Mesozoic tectonic structures and paleogeographic domains.

The review of the historical and instrumental seismicity along the SVFS shows that it does not appear to have generated large earthquakes during the last few hundred years. The moderate seismicity points to a dextral strike-slip activity, which is also corroborated by the field analysis of antithetic Riedel structures of the fault cropping out along the northern segment. Conversely, the southern segment shows geological evidence of sinistral strike-slip activity. The geological and seismological apparently conflicting data can be reconciled considering the faulting style of the southern segment as driven by the indentation of the Adriatic plate, while the opposite style along the northern segment can be explained in a sinistral opening "zipper" model, where intersecting pairs of simultaneously active faults with different sense of shear merge into a single fault system.

# 1 Introduction

Many foreland areas in the world display transverse structures accommodating the segmentation of the thrust fronts and the outward propagation of the fold and thrust belts. These structures are commonly pre-existing, steep normal faults penetrating the basement, and originated during previous episodes of extension at various angles to the belts, often reactivated as transpressional or transtensional shear zones. Given their geometry, during the chain shortening these transverse faults are reactivated as strike-slip faults, as for example in the Himalayan foreland (e.g.: Duvall et al., 2020), in the Laramide belt (e.g.: Bader, 2019), in the Southern Pyrenees foreland (e.g.: Carrillo et al., 2020), in the Northern-Central Apennines (Tavarnelli et al., 2001; Butler et al., 2006; Peacock et al., 2017), in the Southern Apennines foreland (e.g.: Argnani et al., 2009) and in the foreland of the Sicilian-Maghrebian chain (Di Bucci et al., 2006; 2010; Fedorik et al., 2018). In any case, they exert a fundamental tectonic control in basin and thrust belt evolution, but also in gas and oil migration and accumulation (e.g.: Cai and Lu, 2015). The complexity arising from segmentation of the transverse structures has been unravelled also by analogue modelling (Fedorik et al., 2019) and may produce releasing structures with local rising of hot waters (e.g.: Pola et al. 2014a; Torresan et al., 2020).

The Eastern Southern Alps in northern Italy, an ENE-WSW trending fold and thrust retrobelt of the Alpine chain located at the northern boundary of the Adriatic indenter, is cut by the Schio-Vicenza Fault System (SVFS). The northernmost Adriatic indenter is bounded by the Giudicarie fault system to the west, the Pustertal-Gailtal fault to the north (i.e. the Periadriatic Lineament) and the Dinaric fault system to the east (Fig. 1). The SVFS is a complex transverse structure with respect to the general trend of the thrust fronts, and is composed of a set of NW-SE trending and NE-dipping high angle faults, that lowers to the east the sedimentary succession of the northern Adria plate, and it is recognised for a total length of about 150 km from the Adige River valley to the Po River delta (Fig. 1). The main structure of this fault system is the so-called Schio-Vicenza Fault (SVF, in the oldest literature also known as Schio Line or Vicenza Line), a NW-SE, steep and very prominent strike-slip fault. Due to his imprint on the landscape between the towns of Schio and Vicenza (Fig. 1), this fault has been recognized since the beginning of the geological studies of the area, firstly by the Austrian author Schauroth (1855). The SVF is recognised as a prominent morphological lineament that abruptly separates the Lessini, Berici Mountains and Euganei Hills block, interpreted as an "undeformed" foreland structural high (Bigi et al., 1990), from the Veneto-Friuli foreland basin (e.g. Toscani et al., 2016). The SVF has been mapped by some authors also in the mountain area north of Schio up to the Adige valley (e.g. Bittner, 1879; von Klebelsberg, 1918; De Boer, 1963; Semenza, 1974; Zanferrari et al., 1982; Cantelli and Castellarin, 1994; Laubscher, 1996; Castellarin and Cantelli, 2000), and in the plain south of Padova (e.g. Finetti, 1972; Pola et al., 2014b).

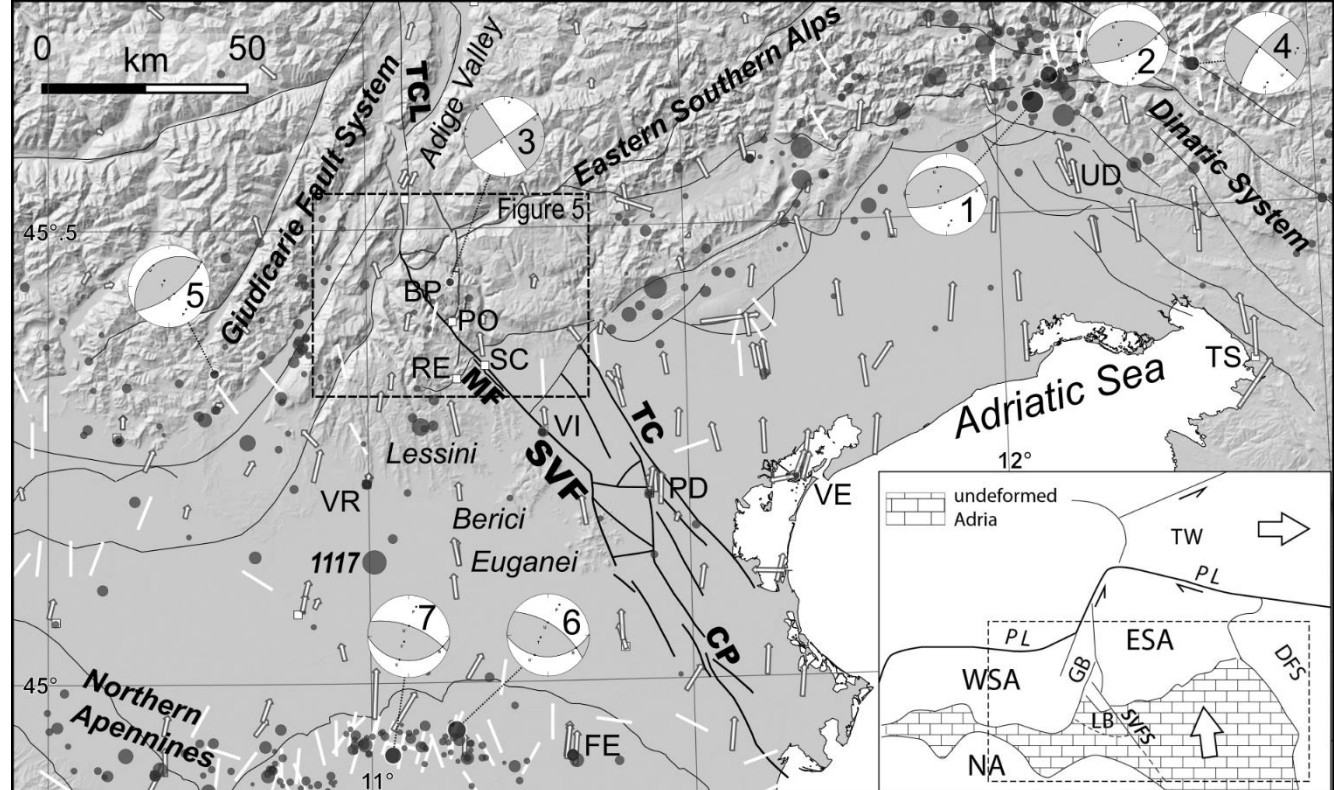

**Figure 1**. Seismotectonic map of northern Italy showing historical and instrumental earthquakes of M>4.5 with grey circles proportional to the magnitude from the CPTI15 (Rovida et al., 2020; 2021) and ISIDe databases (ISIDe Working Group, 2007), focal mechanisms of main events, GPS velocity field (from Devoti et al., 2017; white arrows) and SHmax orientations (from IPSI database, Mariucci and Montone, 2020; white bars). Focal mechanisms (from Pondrelli et al., 2020): (1) $M_w$ 6.4, 6 May 1976 and (2) $M_w$ 6.0, 15 September 1976 Friuli earthquakes; (3) $M_w$ 4.9, 13 September 1989; (4) $M_w$ 5.6, 12 April 1998 Bovec-Krn Mt. earthquake; (5) $M_w$ 5.0, 24 November 2004; (6) $M_w$ 6.1, 20 May 2012 and (7) $M_w$ 5.9, 29 May 2012 Emilia earthquakes. Trace of the faults of the SVFS from Pola et al. (2014a) and Torresan et al. (2020). The inset shows the regional geodynamic framework of the Adria indenter. ESA: Eastern Southern Alps, corresponding to the northern shortened Adria plate margin; GB: Giudicarie Belt; LB: Lessini Block; NA: Northern Apennines outermost front; PL: Periadriatic Lineament; TW: Tauern Window; WSA: Western Southern Alps. Names of the faults: CP: Conselve-Pomposa Fault; MF: Malo Fault; SVF: Schio-Vicenza Fault; TC: Travettore-Codevigo Fault; TCL: Trento-Cles Fault. Cities and geographic names: BP: Borcola Pass; FE: Ferrara; PD: Padova; PO: Posina; RE: Recoaro; SC: Schio; TS: Trieste; UD: Udine; VE: Venezia; VI: Vicenza; VR: Verona. Digital elevation model of central map is EU-DEM v1.1. Available online: https://land.copernicus.eu/imagery-in-situ/eu-dem (accessed on 20 June 2021).

To the south-east of the Euganei Hills, the SVFS continues in the subsurface of the Veneto plain, where it is known as the Conselve-Pomposa fault (CP) (Pola et al., 2014b). Eastward, another buried segment is the Travettore-Codevigo fault (TC; Fig. 1; Pola et al., 2014b).

In the centuries after Schauroth's work only the SVF s.s. has been the subject of many scientific works, and has been cited many times and frequently used in regional tectonic models as a key feature accommodating the indentation of the northern Adria plate margin (e.g. Semenza, 1974; Slejko et al., 1989; Castellarin et al., 2000; Zampieri et al., 2003; Heberer et al., 2017; Mantovani et al., 2020). The work of Pola et al. (2014b) defined for the first time the complex architecture of the whole SVFS. South of Schio, the SVF is the westernmost strand of a set of buried faults which lies in the subsurface of the Quaternary Veneto plain. North of Schio, near the town of Posina, the SVF joins with a set of N-trending conjugate faults.

Given its geometry and orientation, the most recent activity of the SVFS during the shortening of the Southern Alps was dominated by strike-slip movements. However, the bedrock fault planes of the SVFS are poorly exposed in outcrop, so few kinematic indicators can be found. On the other hand, the seismic sections encountering the strands of the SVFS in the Veneto Plain say little about the strike-slip activity, while showing mainly the extensional character of the former tectonic phases (Pola et al., 2014b). Therefore, the recent kinematics of the SVFS remains unclear. Another puzzling character of the SVFS is the

weak seismic activity, which would be expected given the regional role of the fault delimiting the Eastern Southern Alps active and seismogenic thrust fronts (Zanferrari et al., 1982; Galadini et al., 2005; Burrato et al., 2008). Historical and instrumental catalogues show only a scattered distribution of low seismicity along this structure (see Chapter 4). Few instrumental data of the northern part of the area point to a dextral strike-slip activity (e.g. Pondrelli et al., 2006) and consequently, the SVF was recently included in the Italian database of seismogenic source DISS referring to such kinematics (Basili et al., 2008; DISS Working Group, 2018). In spite of this, most of the geological reconstructions refer to a sinistral strike-slip activity (see Table 1).

Since the literature on SVF and SVFS spans a century and a half, we try to summarize data and interpretations flourished in this long interval of time. First, we briefly outline the geodynamic role, the kinematics, the age of the tectonic phase(s) and where possible, the throw and heave derived from the main papers citing the SVF s.s. or the SVFS. We then review the seismological data and finally we try to interpret all the information in a unifying model, which can reconcile apparently contrasting data.

| ID | Year | Authors | Geodynamic role | K | Age of tectonic activity | Throw | Horizontal offset |
|----|------|---------|-----------------|---|--------------------------|-------|-------------------|
| 1 | 1855 | Schauroth | N/A | N/A | N/A | N/A | N/A |
| 2 | 1875 | Suess | N/A | N/A | N/A | N/A | N/A |
| 3 | 1879 | Bittner | N/A | N/A | N/A | N/A | N/A |
| 4 | 1882 | Taramelli | N/A | N/A | N/A | N/A | N/A |
| 5 | 1882 | Molon | Eastern block subsidence | N | Late Tertiary | N/A | N/A |
| 6 | 1901 | Negri | Fault in a map | N/A | N/A | N/A | N/A |
| 7 | 1906 | Maddalena | N/A | 1. N 2. L | N/A | N/A | N/A |
| 8 | 1918 | Klebelsberg | N/A | 1. N 2. L | N/A | 700 m | N/A |
| 9 | 1920 | De Pretto | Eastern block subsidence | 1. N 2. L | Late Miocene Late Miocene | N/A | 2 km (Schio) 16 km (Vicenza) |
| 10 | 1920 | Fabiani | Fault in a map | N/A | N/A | N/A | N/A |
| 11 | 1923 | Pia | N/A | N/A | N/A | N/A | N/A |
| 12 | 1963 | De Boer | Accommodation of magmatism, Orogenic collapse | 1. N 2. L | Early-middle Triassic (N) Early-middle Eocene (N) Middle-late Miocene (L) Early Pliocene (L) | 500- ? m (South) | few kilometers (North) |
| 13 | 1968 | Bosellini et al. | Fault in a map | 1. N 2. L | N/A | N/A | N/A |
| 14 | 1968 | Braga et al. | Fault in a map | 1. N 2. L | N/A | N/A | 2 km |
| 15 | 1972 | Finetti | Eastern block subsidence | N | Quaternary | 320 m | N/A |
| 16 | 1974 | Semenza | Intraplate transfer fault | L | N/A | N/A | Several km |
| 17 | 1982 | Zanferrari et al. | Separation of blocks | 1. N 2. L | Plio-Quaternary | N/A | N/A |
| 18 | 1986 | Sedea, Di Lallo | Fault in a map | L | Neogene-Quaternary | N/A | 3 km |
| 19 | 1987 | Doglioni, Bosellini | Dinaric peripheral bulge | N | Eocene | N/A | N/A |
| 20 | 1988 | Pellegrini | N/A | 1. N 2. L | Triassic (N) Neogene-Quaternary (L) | 500 m | N/A |
| 21 | 1991 | Castaldini, Panizza | Eastern block subsidence | N | Middle Pleistocene-Olocene | 200 m | N/A |

| 22 | 1994 | Cantelli, Castellarin | Transfer fault | L | Messinian-Pleistocene | N/A | N/A |
|----|------|-----------------------|----------------|---|------------------------|-----|-----|
| 23 | 1996 | Laubscher | W margin of clockwise rotating Tauern sublid | L | Middle-late Miocene | N/A | N/A |
| 24 | 2003 | Zampieri et al. | Indenter margin | L | Neogene | N/A | N/A |
| 25 | 2006 | Castellarin et al. | Transfer fault | L | Messinian-Present | N/A | N/A |
| 26 | 2006 | Mantovani et al. | Indenter margin | L | Neogene | N/A | N/A |
| 27 | 2006 | Massironi et al. | Indenter margin | 1. N 2. R 3. L | Paleogene (N) Neogene (R, L) | N/A | N/A |
| 28 | 2008 | Viganò et al. | Strain partitioning within Adria indenter | R | Present | N/A | N/A |
| 29 | 2012 | Fondriest et al. | N/A | R | Neogene-Present | N/A | N/A |
| 30 | 2012 | Masetti et al. | Eastern block subsidence | N | Carnian to Pleistocene | N/A | N/A |
| 31 | 2014a | Pola et al. | Indenter margin | L | Neogene-Present | 0 - 430m | N/A |
| 32 | 2014 b | Pola et al. | 1. Passive margin 2. Indenter margin | 1. N 2. L | Mesozoic (N) Plio-Quaternary (L) | N/A | N/A |
| 33 | 2014 | Turrini et al. | Wrench-type swarm | N/A | Post-Pliocene | N/A | N/A |
| 34 | 2015 | Vannoli et al. | Transverse structure | R | Mesozoic Plio-Quaternary | N/A | N/A |
| 35 | 2016 | Serpelloni et al. | Transfer fault | L | Late Miocene-Quaternary | N/A | N/A |
| 36 | 2017 | Heberer et al. | Transfer fault | L | Middle Miocene-Present | N/A | N/A |
| 37 | 2019 | Brancolini et al. | 1. Passive margin 2. Indenter margin | 1. N 2. L | Mesozoic (N) Plio-Quaternary (L) | N/A | N/A |
| 38 | 2021 | Verwater et al. | Transfer fault | L | Middle Miocene-Present | N/A | 4 km |

**Table 1**: Geological review of more than a century and a half of literature, and synoptic view of the main kinematic parameters of the SVF as proposed by various investigators. At least one third of the authors suggest that a change in kinematics has occurred over time. K: kinematics; N: normal; L: left-lateral; R: right-lateral; N/A: data not available.

**2 More than a century and a half of studies**

Being the most prominent feature south of the Alps in the Veneto plain (Fig. 1), the SVF was drawn in all the geologic and structural maps of northeast Italy at various scales. Therefore, the knowledge of the structure is deeply rooted in the geologic literature and spans for more than a century and a half (Table 1). While few papers have investigated in detail the structure or part of it, the list of works citing the SVF is very long. Here we shortly examine the main papers, books and maps dealing with the SVF with the aim of presenting the evolution of its interpretation, although not giving an exhaustive list. Only recently, due to the availability of subsurface geophysical data, the analysis was extended to buried strands of the structure, which consequently was recognized as a complex system of faults (SVFS). As a consequence, we will use throughout this section both SVF and SVFS.

The 19th century geological school of Vienna produced several sound earth scientists studying the Südtirol region, presently the Trentino-Alto Adige autonomous region of Italy. They extended the research to the conterminous Veneto region, which belonged to Italy during the Habsburg Monarchy (1804-1867) and the Austro-Hungarian Empire (1867-1918). It has been

suspected that these geologists were also military officers with the purpose of knowing the border territories (Scalia, 1917). In fact, although they reported thorough publications on the stratigraphy, palaeontology and tectonics of the Italian territories lying at the southern Austrian border, their geological maps never represented the Austrian side. The first mention of the SVF appeared in the work of the Austrian author Julius Schauroth (1855) on the geological setting of the Recoaro area (Eastern Southern Alps).

Eduard Suess (1875) at page 33 of his book "On the formation of the Alps" marked the discontinuity of the Tertiary formations at the foot of the southern margin of the Alps in between the towns of Schio and Vicenza. The short citation of the SVF is contained in just a sentence, deriving from the geomorphological prominence of the feature, which has later attracted the attention of a number of geologists. Another Austrian author, Alexander Bittner (1879), suggested that the SVF does not end at Schio, a village lying at the foothills of the Pre-Alps, but was extending within the chain until at least the Borcola pass, 18 km northwestwards. Later-on in the 20th century the Italian geologists inherited the plentiful literature on the Southern Alps published in the German language. The SVF was then depicted in the geological map of Negri (1901) or referred to as a fault separating abruptly the mountain from the plain, thus with a downthrow of the eastern block (Taramelli, 1882; Molon,1882; Maddalena, 1906). The last author mapped the SVF southeastwards till the Euganei Hills relating their magmatism to the fault, and for the first time recognized also a left-lateral strike-slip movement. The Austrian geologist von Klebelsberg (1918) dedicated his paper to the thesis of the northwestern extension of the fault until the Adige Valley, furthermore to the north than the Bittner (1879) proposal. In addition to vertical offsets of the fault, he also recognized strike-slip movements. In contrast, Fabiani (1925) in his geological map recognised the SVF northern tip near the Borcola Pass.

The paper from the Italian geologist De Pretto (1920) added an estimation of the amount of the strike-slip component, showing an important gradient of the offset from 16 km in the southeastern portion (close to Vicenza) to only 2 km close to Schio.

In his monograph on the tectonics of the Lessini Mts., the Austrian geologist Julius Pia (1923) drew an about 25 km long SVF ending at Posina, i.e. 10 km northwest of Schio. Forty years later the Dutch geologist Jelle De Boer (1963) published his doctoral thesis in the Vicentinian Pre-Alps under the supervision of van Bemmelen. He referred to the SVF fault as the Vicenza fault, a regional structure extended for over 100 km from the Adige valley till at least the Euganei Hills to the southeast. In between the towns of Schio and Vicenza, he distinguished in the footwall of the SVF another parallel fault segment named Malo Fault (MF; Fig. 1). Two components of movement were recognized: a vertical one, with a throw of ca. 500 m north of Schio, but important even in the southern part, where the fault was described as a normal fault, and a strike-slip component, which predominates in the northern sector, where the fault was described as a left-lateral wrench fault. De Boer (1963) unravelled a very complicated history of the SVF, with at least four tectogenetic phases. In the first two, the structure acted as a normal fault, accommodating vertical movements produced by the intrusion of respectively felsic magmas, during the Scythian and Anisian, and mafic magmas during the Eocene. A switch of the dip from west to east occurred shifting from the northern sector to the southern sector. In the last two phases of activity, the SVF acted as a left-lateral strike-slip fault in the early-middle Miocene and in the early Pliocene. This kinematics has been related to differential movements of the eastern and western blocks of the fault, in conjunction with gravitational collapse of the sedimentary cover from the Recoaro horst.

From the De Boer work (1963) onwards the literature on the SVF is almost exclusively by the Italian authors. In the sixties the national project on geological cartography at a scale 1:100,000 was in full operation. Therefore, in the sheets 49 *Verona* (Bosellini et al., 1968) and 36 *Schio* (Braga et al., 1968), the SVF and the MF were drawn according to the previous literature. Studying the plain region between the Venice lagoon and the Euganei Hills using new seismic lines, Finetti (1972) drew the southern portion of the SVF as a very steep structure with a Quaternary throw of 200-320 m. The lowering of the eastern block would be ongoing in the context of the subsidence of the north Adriatic region started since the Eocene.

A regional study from Semenza (1974) attributed to the SVF an important role in accommodation of differential shortening of the two sides of the Italo-Dinaric (now Adria) plate, which was divided into two parts by the structure extending from Cles

(north of Trento) to the Dalmatia in the eastern coast of the Adriatic Sea. The left-lateral movement of the SVF was explained by the presence in the western block of rocks deforming in a "ductile" way sandwiched between the "stiff" Adamello pluton and the Athesian platform (the Jurassic Trento platform, partly coinciding with the thick Permian volcanic district). The absence in the eastern block of such soft and rigid rocks would have permitted the northwestwards slip of the eastern side of the SVF.

In the late seventies to early eighties the SVF was investigated in the context of the "C.N.R. Finalized Project Geodynamics - Subproject Neotectonics" (Barbieri et al., 1981; Zanferrari et al., 1982). The work of Zanferrari et al. (1982) concerning the neotectonic evolution of north-eastern Italy, investigated five paleogeographic reconstructions of eight units, from early Pliocene to the Present. For the unit corresponding to the eastern Veneto and western Friuli plains, i.e. the foreland basin of both the Dinarides and the Eastern Southern Alps, the activity of NW-SE faults bounding sub-basins has been considered. The SVF is traced for more than 100 km from the Adige valley to the Po river and representing the longest fault in north-eastern Italy. In the early Pliocene the SVF was separating the stable continental area of the Lessini Mountains from a gulf with marine sedimentation. In the late Pliocene the eastern block was rising and the marine environment became a continental area. In this period, a left-lateral strike-slip component of the fault movement started. In the early Pleistocene the sea re-entered into the gulf and the fault was active as a consequence of the differential uplift of the Lessini Mountains area and the Veneto plain. Left-lateral strike-slip movements are also detected. In the mid-late Pleistocene, the SVF was again active separating the uplifting Lessini Mountains area from the subsiding plain. Finally, Zanferrari et al. (1982) recognized that in the final Pleistocene to Holocene the SVF accommodated only differential vertical movements of the Lessini Mountains with respect to the subsiding plain.

The explanation text of the geological map at a scale 1:10,000 of the Pasubio-Posina-Laghi area from Sedea and Di Lallo (1986) recognizes a Neogene to Quaternary sinistral strike-slip activity of the SVF, with a minimum offset of 3 km.

An Eocene (Lutetian) origin of the SVF was proposed by Doglioni and Bosellini (1987). In two sketches, the fault was represented as a normal fault of a set of ENE dipping faults developed in the eastern margin of the forebulge of the WSW verging Dinaric fold-and-thrust belt.

The title of a paper from Pellegrini (1988) relies directly on the morphological and neotectonics evidence of the SVF. On the basis of fluvial terraces nesting and stream deviation close to Schio, the author argues that the eastern block of the fault (the plain in between Schio and Vicenza) was lowered during the Quaternary.

The research for the C.N.R. Finalized Project Geodynamics - Subproject Neotectonics led Castaldini and Panizza (1991) to propose an inventory of active faults in between the Po and Piave rivers, where the SVF was classified as "active" within the class "II degree of activity", i.e. with an average slip rate ranging from 0.1 to 1 mm/year, corresponding to a relative lowering of the eastern block in the plain sector. The strike-slip component of the mountainous sector remained controversial. The only portion of this very long structure showing features corresponding to a "I degree of activity" (average slip rate ranging from 1 to 10 mm/year) was referred to as the Malo fault (MF in Fig. 1), a ca. 15 km-long synthetic segment outcropping between Schio and Vicenza.

Cantelli and Castellarin (1994) have interpreted the SVF as a sinistral transfer fault separating the central Southern Alps, whose compressional deformation stopped during the Tortonian, from the Eastern Southern Alps, whose deformation is still active.

An interesting although complicated model has been proposed by Laubscher (1996), who rejects the model of the northern Adria indentation against the Eastern Alps (e.g. Ratschbacher et al., 1991). The SVF is interpreted as a boundary of the Veneto foreland, delimiting the structural high called "Adige embayment" (the Lessini Mountains block), separating the Central from the Eastern Southern Alps, which is defined kinematically "irksome". This author envisaged a SVF with a sinistral strike-slip faulting style, separating to the west the clockwise rotating Tauern sublid (the crust below the brittle-ductile transition) during

the Jura phase (middle-late Miocene). These transcurrent and rotational movements were interpreted to be induced by the NW-SE Adria-Europe convergence along with deep-seated anticlockwise rotation of Adria.

The analysis of the Neogene linkage of a set of sinistral strike-slip, N-S inherited Permian to Cretaceous extensional faults conjugate with the SVF has been performed by Zampieri et al. (2003). In this study, the strike-slip kinematics of the SVF north of Posina, where the junction with a set of linked inherited fault occurs, was recognized to have been both dextral (in the Paleogene) and sinistral (in the Neogene), according to the development of a lens-shaped pop-up. This contractional feature has been related to the dextral bend of a sinistral SVF northwards linking with the N-S Trento-Cles fault (TCL in Fig. 1), already suggested by Semenza (1974).

Within the neo-Alpine evolution of the Southern Eastern Alps, Castellarin and Cantelli (2000) interpreted the Schio-Vicenza structural system as a sinistral transfer fault system between the Giudicarie belt (inner part of the chain) and the Pedemontana (frontal structure). This contractional deformation (Adriatic compressional event with shortening axis trending WNW-ESE), would be younger than the Serravallian-Tortonian Valsugana event (shortening axis NNW-SSE) and considered mainly Messinian-Pliocene in age.

An extensive regional examination of the SVF by means of structural, geochronological and seismotectonic data has been performed by Massironi et al. (2006). The post-Oligocene kinematic linkage of inherited Permian to Mesozoic structures at the western boundary of the north Adriatic (Dolomites) indenter would have originated a sinistral strike-slip mega shear zone, of which the SVF is a key element, being an incipient divide between the nearly stationary westernmost part of the North Adriatic indenter and the still northward-pushing main body of the Adria plate.

The Neogene to Present sinistral strike-slip faulting style of the SVF has been attributed once again to its role of western rail of the north Adriatic indenter by Mantovani et al. (2006). This kinematics would have accommodated the decoupling of the Adriatic block from its northwestern (Padanian) protuberance and from Africa, induced by the late Messinian counterclockwise rotation of the Adria plate.

The computation of the stress and strain from relocated seismic events in the Giudicarie and Lessini Mountains regions of the Southern Alps permitted Viganò et al. (2008) to define the right-lateral strike-slip activity of the NW-trending faults (parallel to the SVFS). It is worth mentioning that the analyzed events of the Lessini Mountains fall in the northern sector of the SW block of the SVF (i.e. footwall block; Lessini Mountains), where the shortening axis is ca. N-S, while in the NE block (Eastern Southern Alps foreland) the shortening axis is NNW-trending. Moreover, in a subsequent paper (Viganò et al., 2015), which include the crustal rheology analysis, the location of the seismicity in the northern part of the Lessini Mountains has been related to the Verona-Vicenza gravity high anomaly (intrusive and effusive volcanic rocks), which contribute to a stronger lithosphere.

An outcrop of an exhumed ca N-S trending sub-vertical strike-slip fault in the damage zone of the SVF close to the Borcola Pass (7 Km north of Posina) has been studied by Fondriest et al. (2012). This conjugate fault cuts the dolostones of Upper Triassic age and the macroscopic and microscopic analysis of secondary fracture patterns suggests a sinistral shear sense. This conclusion suggests that in this area the s.s. SVF is dextral.

By means of integration of stratigraphic and geophysical data, Masetti et al. (2012) have reconstructed the extensional architecture of the Alpine foreland subsurface. Their interpretation of a seismic section at high angle to the SVF south of Padova permits to recognize the throw component of the fault, which presents offsets of the pre-Upper Messinian unconformity units with similar values, but a minor offset of the base of the Pleistocene.

Close to Padova the Euganean hydrothermal system, one of the most important water-dominated low-temperature geothermal systems in Europe (Fabbri et al., 2017), presents the outflow of the hot waters in correspondence of the SVFS, suggesting a structural control. A multidisciplinary study including the analysis of the fracture pattern of a travertine mound, relict of the main outflow which was active until the middle of the 20th century, before a large exploitation for tourism purposes, has

permitted to infer in the subsurface the existence of an extensional relay zone linking two main fault segments of the SVFS (Fig. 1; Pola et al., 2014a, 2015). The architecture of the western margin of the North Adriatic foreland has been investigated by Pola et al. (2014b) by means of some seismic lines. This margin is characterized by a set of mainly blind structures belonging to the SVFS. While the vertical component of the offsets is very clear on the seismic lines, pointing to a Mesozoic extensional deformation of the eastern border of the Trento platform, particularly for the eastern buried segments of the system, the Neogene to Present strike-slip component remains puzzling.

The structural model of the Euganean geothermal system has been refined by Torresan et al. (2020) by means of a numerical 3D reconstruction of the subsurface structure. In this work for the first time the deep part of the faults of the SVFS is shown. A 3D structural model of the Po valley performed by Turrini et al. (2014) shows clearly in several figures the presence of the SVFS, which is simply referred to as a wrench-type fault swarm.

In the context of the seismotectonics of the Po Plain, Vannoli et al. (2015) have subdivided the seismogenic sources in four groups of mainly blind faults. The SVF was included in the "Transverse structures" group, a set of faults roughly perpendicular to the chains bordering the Po Plain (Northern Apennines and Southern Alps). These structures would be the brittle response of the upper lithosphere to the lateral changes in the geometry of the Adriatic subducting slab, being characterized by relatively deep earthquakes. These investigators proposed that a fault segment belonging to the northern portion of the SVFS could be the source of the 13 September 1989 earthquake ($M_w$ 4.9, depth 9 km, dextral strike-slip faulting; ID 3 in Fig. 1). Accordingly, in the national Database of Individual Seismogenic Sources 3.2.0 version (DISS Working Group, 2018) this NW-trending fault has been classified as dextral.

Serpelloni et al. (2016) using a kinematic modeling of geodetic data, calculated the recent slip rate of the SVF, distinguished in a southern sector bordering the Veneto foreland and characterised by a left-lateral slip-rate of $0.4 \pm 0.2$ mm/yr, and a northern one with a right-lateral a slip-rate of $1.0 \pm 0.1$ mm/yr.

The role of the SVF as sinistral strike-slip western rail of the northern Adriatic indenter has been recognized also by Heberer et al. (2017), starting from the latest Mid-Miocene, and by Brancolini et al. (2019), after the Early Pliocene.

Finally, Verwater et al. (2021) have performed a tectonic balancing of a network of seven cross sections parallel to the shortening direction of the Eastern Southern Alps across the transition between two structural domains of the Mesozoic Trento Platform. This sharp transition is referred to the sinistral strike-slip Schio-Vicenza-Trento-Cles fault system, which accommodated different amount of shortening due to lateral variation in strength of the Adriatic indenter, related to Permian - Mesozoic tectonic structures and paleogeographic domains.

## 3 Geological evidence

In this section we present selected geological and geophysical features along the SVFS (Fig. 2). North of Schio, i.e. north of the Pedemontana thrust (PE in Figs 1, 5 and 6) , the SVF does not outcrop anywhere, but geomorphic features such as straight valleys and saddles are aligned along the NW direction. At the Borcola Pass, two inactive quarries have produced the exposure of vertical fault planes trending ca N-S, with prominent strike-slip striations (IDs 1 and 2; Fig. 2; Table 2). These faults belong to the damage zone of the SVF, which is 2-3 km wide (Fondriest et al., 2012) and are interpreted as antithetic Riedel shear fractures (Zampieri et al., 2003). Since their kinematic indicators are sinistral, in this section the SVF is inferred to be dextral.

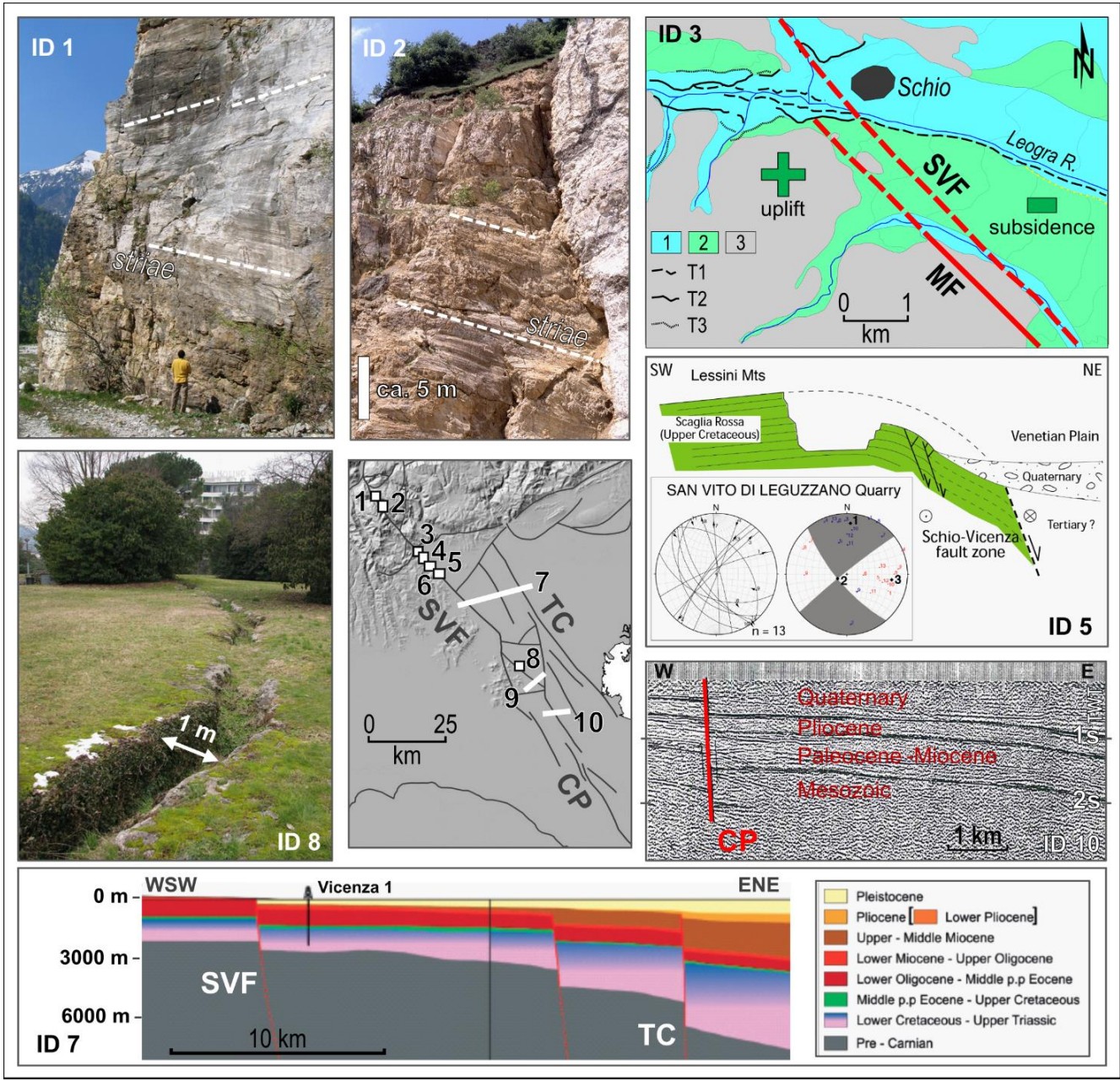

**Figure 2**. Geological and seismic evidence supporting the activity and style of faulting of the SVFS (see Table 2 for details, and central inset for locations). ID 1: vertical fault plane with strike-slip lineations (Borcoletta quarry); ID 2: vertical fault plane with strike-slip lineations (Borcola quarry); ID 3: river terraces (T1, T2 and T3) crossing the SVF showing the differential uplift of the two blocks separated by the fault (redrawn from Pellegrini, 1988); Symbols: 1, Holocene deposits, 2, Pleistocene deposits, 3, pre-Quaternary bedrock; ID 5: cross section showing a drag fold deforming the Upper Cretaceous Scaglia Rossa formation, where the mesofault inversion suggests that the last activity was sinistral strike-slip with a kinematic shortening axis oriented nearly east-west (adapted from Zampieri et al., 2009); ID 7: interpreted seismic lines orthogonal to the SVFS (adapted from Pola et al., 2014b); ID 8: Abano Terme, main fracture of the Montirone travertine mound (Pola et al., 2014a); ID 10: seismic line orthogonal to the CP fault (adapted from Finetti, 1972). The IDs 4 and 6 can be found in the sheets 36 *Schio* (Braga et al., 1968) and 49 *Verona* (Bosellini et al., 1968) of the 1: 100,000 scale Geological Map of Italy. See also Table 2. Names of faults as in Figure 1. Digital elevation model of central map is EU-DEM v1.1. Available online: https://land.copernicus.eu/imagery-in-situ/eu-dem (accessed on 20 June 2021).

| ID | Locality | Type | Fault | Age | K | Reference(s) |
|----|----------|------|-------|-----|---|--------------|
| 1 | Borcoletta quarry | - Geological sections<br>- Inversion of fault-slip data | RSVF | Neogene | L | Zampieri et al., 2003<br>Fondriest et al., 2012 |
| 2 | Passo Borcola quarry | Inversion of fault-slip data | RSVF | Neogene | L | Zampieri et al., 2003<br>Fondriest et al., 2012 |
| 3 | Torrente Leogra | Differential uplift registered by alluvial terraces | SVF | Pleistocene | N | Zanferrari et al., 1982<br>Pellegrini, 1988 |

| 4 | Magrè | Geological structure | MF | Neogene (Post-Upper Oligocene) | N | Braga et al.,1968 Zanferrari et al., 1982 |
|---|---|---|---|---|---|---|
| 5 | San Vito di Leguzzano | - Geological structure (drag fold) - Inversion of fault-slip data | SVF | Post-Cretaceous | 1. N 2. L | Zanferrari et al., 1982 Zampieri et al., 2009 |
| 6 | Isola Vicentina | Geological structure (drag fold) | SVF | Neogene (Post-Aquitanian) | N | Bosellini et al., 1968 Zanferrari et al., 1982 |
| 7 | Vicenza | Seismic section | SVF | Pleistocene | N | Pola et al., 2014b |
| 8 | Abano Terme | Fractures of a travertine mound | SVF | < 20 Ky | L | Pola et al., 2014a |
| 9 | Battaglia Terme | Seismic section | SVF | Pleistocene | N | Masetti et al., 2012 |
| 10 | Conselve | Seismic section | CPF | Pliocene | N | Finetti, 1972 Pola et al., 2014b |

**Table 2**: Field geological evidence and seismic data of the SVFS as reported in literature. The IDs 1 and 2 are located north of the Posina Triple Junction and are antithetic Riedel structures respect to the SVF, while the other IDs are to the south. SVF: Schio-Vicenza Fault; RSVF: Antithetic Riedel of the SVF; MF: Malo Fault; CPF: Conselve-Pomposa Fault. Column labelled with K refers to the kinematic interpretation: N, normal fault; L, left-lateral; R, right-lateral.

The analysis of the longitudinal profiles of the fluvial terraces close to Schio shows Pleistocene differential uplift across the fault system, with the western block (footwall) of the SVF uplifted with respect to the plain (hanging-wall) and accordingly, showing a higher number of river terraces (ID 3; Fig. 2 and Table 2; Pellegrini, 1988).

To the southeast of the intersection with the frontal thrust of the Eastern Southern Alps, from Schio to Vicenza, the SVF has a prominent surficial expression, since it marks the transition from the eastern border of the Lessini Mountains block to the foreland basin plain, which is lowered. In between the SVF and the parallel Malo fault (MF), Miocene sediments dip towards the NE, while in some sectors the extensional displacement of the SVF is testified by the presence of drag folds in the footwall (ID 5 in Fig. 2; IDs 4, 5 and 6 in Table 2; for their location see the central map of Fig. 2). The IDs 4 and 6 can be found in the sheets 36 *Schio* (Braga et al., 1968) and 49 *Verona* (Bosellini et al., 1968), respectively, of the 1:100,000 scale Geological Map of Italy (both available from the website of the Italian Geological Survey: http://sgi.isprambiente.it/geologia100k/; last accessed on 27 May 2021). Some seismic sections crossing the SVFS at a high angle in the plain between Vicenza and Adria (Finetti, 1972; Pola et al., 2014b), two of which are presented in Fig. 2 (IDs 7 and 10), highlight the extensional displacement. The total throw of the different sub parallel segments of the SVFS south of Schio is the cumulative expression of the deformation, started from the Mesozoic time in an extensional stress field close to the eastern margin of the Trento platform, and continues to this day in a contractional stress field connected to the flexuring of the foreland (Zanferrari et al., 1982; Pola et al., 2014b).

Obviously, the strike-slip component of the SVF cannot be pointed out by seismic sections but is registered by the analysis of few mesofaults presented in Fig. 2 (IDs 1 and 2) and extracted from the offset of the sedimentary and volcanic formations in geological maps. In the southern section of the SVF, close to the Euganei Hills, a buried relay ramp, kinematically linking the SVF and the CP, has been inferred from the analysis of stratigraphic logs, of gravimetric maps and of the fracture pattern of travertine deposits (Pola et al., 2014a, 2015). The travertine mound was built by thermal hot water springs, active until the intense exploitation of the second half of the 20th century (Fabbri et al. 2017). Conceptual and numerical modelling of the Euganean geothermal system have corroborated the impact of the structural process driving a local increase in convection and the rising of thermal waters (Pola et al., 2020).

**4 Seismotectonic evidence**

Historical and instrumental seismicity in the NE Italy is distributed along the Giudicarie Fault System, the Eastern Southern Alps fronts and the Northern Apennines fronts (e.g. Rovida et al., 2021). The Eastern Southern Alps thrust fronts are bordered to the east by the dextral strike-slip faults of the Dinaric system (Fig. 1), that hosts the seismogenic sources of a number of strong and moderate events that struck the area both in historical and instrumental times, like the 1511 Friuli-Slovenia, $M_w$ 6.3 and the 1998 Bovec-Krn Mt., $M_w$ 5.6, earthquakes (the latter, ID 4 in Fig. 1; Kastelic et al., 2013; DISS Working Group, 2018). Several destructive earthquakes hit Northern Italy in historical and recent times, and the strongest instrumental earthquakes occurred in Friuli during the 1976 sequence (having a maximum magnitude of 6.4) and in Emilia during the 2012 sequence (having a maximum magnitude of 6.1). The causative faults of the 1976 Friuli sequence are S-directed blind thrusts of the Eastern Southern Alps (IDs 1 and 2 in Fig. 1; e.g. Burrato et al., 2008) while the causative sources of the 2012 Emilia sequence are the most advanced and buried fronts of the Northern Apennines (IDs 6 and 7 in Fig. 1; e.g. Burrato et al., 2012). Both these destructive sequences are characterized by multiple similarly large mainshocks and relatively shallow earthquakes (Vannoli et al., 2015).

The buried outer arc of the Central Southern Alps is connected to the east with the NNE-trending Giudicarie system, that developed along the margin between the Lombardian basin to the west and the Trento platform to the east. This structural system is commonly interpreted as a regional transfer zone between the Central and Eastern Southern Alps (e.g. Castellarin and Cantelli, 2000), and thought to be responsible for the 24 November 2004, $M_w$ 5.0, earthquake (ID 5 in Fig. 1; Pessina et al., 2006) and the 30 October 1901, $M_w$ 5.4, event (Vannoli et al., 2015; DISS Working Group, 2018).

The instrumental earthquakes of northern Italy and Slovenia show prevailing compressional focal mechanisms, confirming that the compression is active and is accommodated by thrust and strike-slip faulting, in agreement with geologic, morphotectonic and paleoseismological evidence (e.g. Galadini et al., 2005; Burrato et al., 2008; Falcucci et al., 2018; Patricelli and Poli, 2020; Viscolani et al., 2020) and instrumental active strain (GPS) data (e.g. Devoti et al., 2017; Anderlini et al., 2020; Fig. 1).

The largest known Po Plain earthquake, the 3 January 1117, $M_w$ 6.7, earthquake (labelled with the year of occurrence in Fig. 1; Guidoboni et al., 2018; Rovida et al., 2021), occurred on an elusive blind fault, in the plain south of Verona, cutting through the shortened foreland of both the Southern Alps and the Northern Apennines. As a matter of fact, the 1117 event hit a portion of the Po Plain to the south of the Lessini Mountains generally considered undeformed and far from the outermost thrust fronts of the two chains.

Besides the relatively well-known shallow thrust sources of the Southern Alps and Northern Apennines, the basement rocks of the Po Plain are cut by deeper inherited faults, that formed during the Mesozoic extensional phases preceding the inception of the Africa-Eurasia relative convergence (e.g. Scardia et al., 2015). As testified by the 1117 and the 13 January 1909, $M_w$ 5.4, events, significant earthquakes may be generated by this set of elusive, inherited structures, that are reactivated in the present-day tectonic compressional regime as reverse or strike-slip faults (Vannoli et al., 2015). In particular, the 1909 earthquake had a macroseismic epicenter located south of Ferrara and affected a very large area with slight to moderate damages (Rovida et al., 2021). This macroseismic scenario was interpreted as caused by the significant depth of the event (Meloni and Molin, 1987; Faccioli, 2013; Vannoli et al., 2015), and, as suggested by Sbarra et al. (2019), it may have occurred well below the basal detachment of the Northern Apennines outer fronts at a depth of 41 km, with a recalculated $M_w$ of 6.2.

According to these interpretations, as a consequence of the paleogeographic and structural inheritance, long-lived faults hosted in the basement rocks, that probably bounded the Mesozoic rift systems, are reactivated in the current stress regime, and are the sources of the most dangerous earthquakes of the Po Plain and surrounding areas, having magnitude comparable or larger to those associated with the active thrusts (e.g. Scardia et al., 2015).

According to the historical seismic catalogues (Guidoboni et al., 2018, 2019; Rovida et al., 2021), during the last millennium the SVFS did not generate earthquakes larger than magnitude 5.0. As a matter of fact, the cities of Vicenza and Padova, which

are located very close to the SVFS (Fig. 1), have a long and well documented "seismic history", i.e. list of effects in terms of macroseismic intensities that affected the two localities, with well-distributed observations over the last millennium, both starting with the 1117 earthquake, even if they have rarely experienced damage due to large earthquakes (Rovida et al., 2020, 2021; Fig. 3). Both the ancient chroniclers and the Benedictine monks of the various monasteries in the area have documented the occurrence of seismic events over the centuries and provided detailed descriptions of the earthquake effects on the cities

and the environment. The great Italian tradition and expertise in historical seismology made possible an interpretation of this wealth of data and a reliable quantitative processing of qualitative historical data (see Guidoboni et al., 2018, 2019; Tarabusi et al., 2020 and references therein).

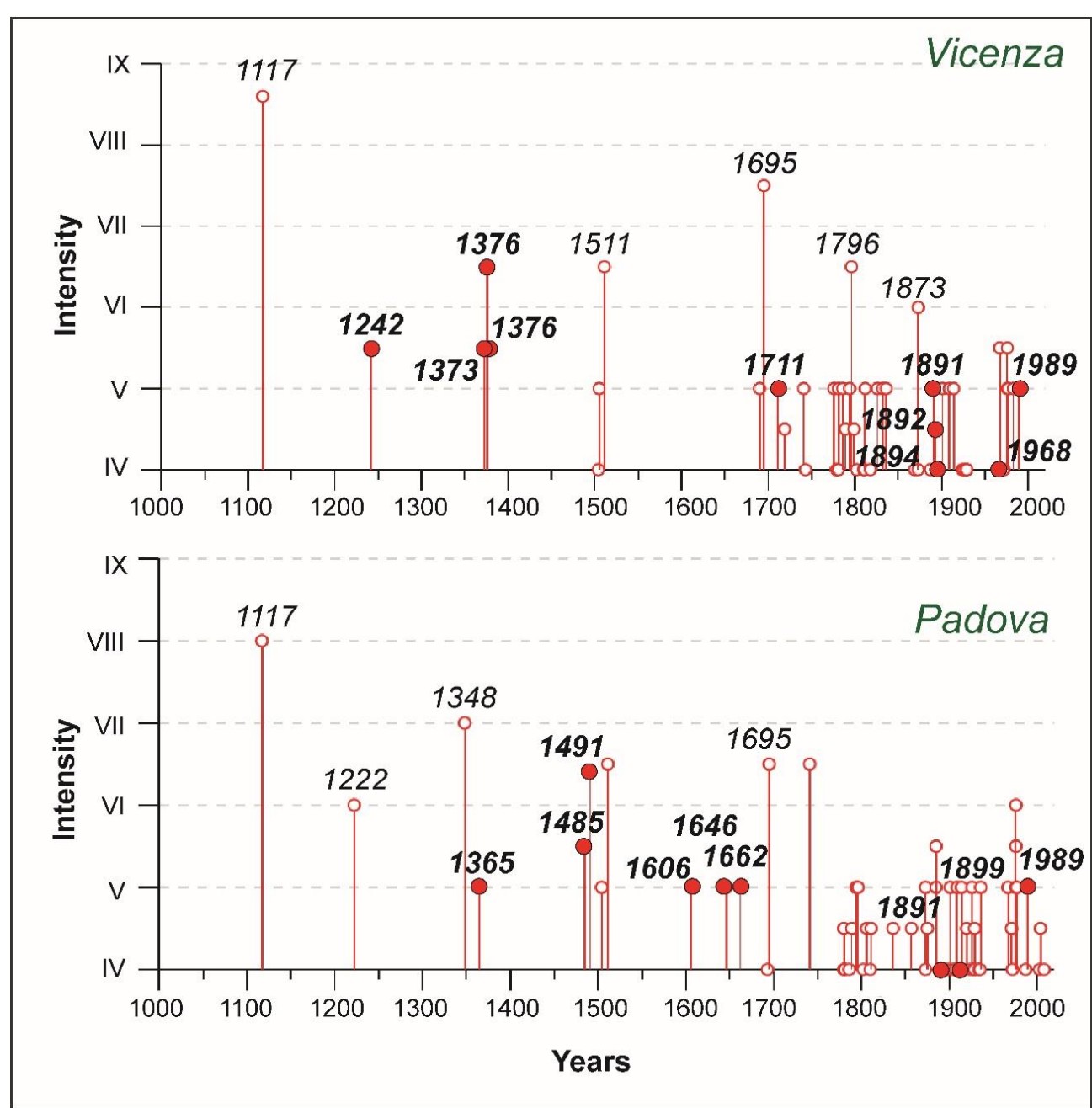

**Figure 3**. Maximum macroseismic intensities recorded in the cities of Vicenza (top) and Padova (bottom) from the year 1000
to today (Rovida et al., 2021). Note that not all the earthquakes shown in figure are listed in Table 3 or Table 4, where only events located in the study area are included. The local earthquakes (i.e. located close to the towns) are labeled in bold and filled by a red circle (see text for detail). The SVFS is the only known active fault system in the area, with a maximum distance of 5-10 km west of the two towns, and for this reason we suggest that presumably have caused these local earthquakes.

The known earthquakes responsible for the most serious damage in the study area were the 1117, $M_w$ 6.5, *Veronese*, the 1348,
$M_w$ 6.6, *Alpi Giulie*, and the 1695 $M_w$ 6.4, *Asolano* events (Fig. 3); their causative faults are far from the SVFS, and do not belong to it. However, numerous historical earthquakes were felt exclusively in Vicenza or Padova, and most likely were generated by local sources (i.e. located close to the towns), potentially belonging to the SVFS that runs few km to the west of the two towns (Fig. 4a). These low-magnitude historical earthquakes, listed and marked with an "X" in the last column of the Table 3 and labeled in bold in Fig. 3, were very often only strongly felt (corresponding to the degree V of the MCS scale).
Some of these local events hit the city of Vicenza between the XIII and XIV centuries (earthquakes ID 1, 3, 4, 5 and 6 in Table 3) and the city of Padova between the XV and XVII centuries (earthquakes ID 7, 8, 9, 10 and 11 in Table 3). Despite the many uncertainties in this regard, these observations may be evidence that small magnitude seismic sequences migrated from north (Vicenza) to south (Padova) along the SVFS over the course of those five centuries.


| ID | Date | Locality | Intensity | N | $M_w$ | SVFS |
|----|------|----------|-----------|---|-------|------|
| 1 | 24 Oct 1242 | Vicenza | V-VI | 1 | 4.4 | X |
| 2 | 04 Mar 1365 | Pianura Veneta | V | 4 | 4.6 | X |
| 3 | Jan 1373 | Vicenza | V-VI | 1 | 4.4 | X |
| 4 | Apr 1373 | Vicenza | V-VI | 1 | 4.4 | X |
| 5 | 12 Mar 1376 | Vicenza | VI-VII | 1 | 4.9 | X |
| 6 | 15 Mar 1376 | Vicenza | V-VI | 1 | 4.4 | X |
| 7 | 1 Sep 1485 | Pianura Padano-Veneta | V | 4 | 4.2 | - |
| 8 | 24 Jan 1491 | Padova | VI-VII | 1 | 4.9 | X |
| 9 | 12 Dec 1606 | Padova | V | 1 | 4.2 | X |
| 10 | 22 Feb 1646 | Padova | V | 1 | 4.2 | X |
| 11 | 29 Dec 1662 | Padova | V | 1 | 4.2 | X |
| 12 | 21 May 1711 | Vicentino | IV | 2 | 3.7 | X |

| | | | | | | |
|---|---|---|---|---|---|---|
| 13 | 07 Jun 1891 | Valle d'Illasi | VIII-IX | 403 | 5.9 | - |
| 14 | 09 Aug 1892 | Valle d'Alpone | VI-VII | 160 | 4.9 | - |
| 15 | 09 Feb 1894 | Valle d'Illasi | VI | 116 | 4.7 | - |
| 16 | 27 Jan 1897 | Prealpi Vicentine | IV-V | 16 | 4.1 | X |
| 17 | 30 Oct 1899 | Prealpi Vicentine | V | 8 | 4.3 | X |
| 18 | 20 Feb 1956 | Padovano | V-VI | 23 | 4.5 | X |

**Table 3**: Pre-WWSSN (World-Wide Standardized Seismograph Network) earthquakes that are located in the SVFS area and were felt in the cities of Vicenza and Padova from the year 1000 to the present day (from CPTI15 catalogue, Rovida et al., 2021, 2020). The earthquakes included in this table are labelled in bold in Fig. 3. The table shows their macroseismic intensities, total number of available macroseismic data points (N), and moment magnitude ($M_w$). Historical earthquakes whose causative faults could presumably be the SVFS are shown with a "X" in the last column; a dash in this column indicates that no association has been made. The 1117 earthquake that is associated with the highest MCS intensity in Padova and Vicenza is not included in this table because its epicenter is located far away from the SVFS.


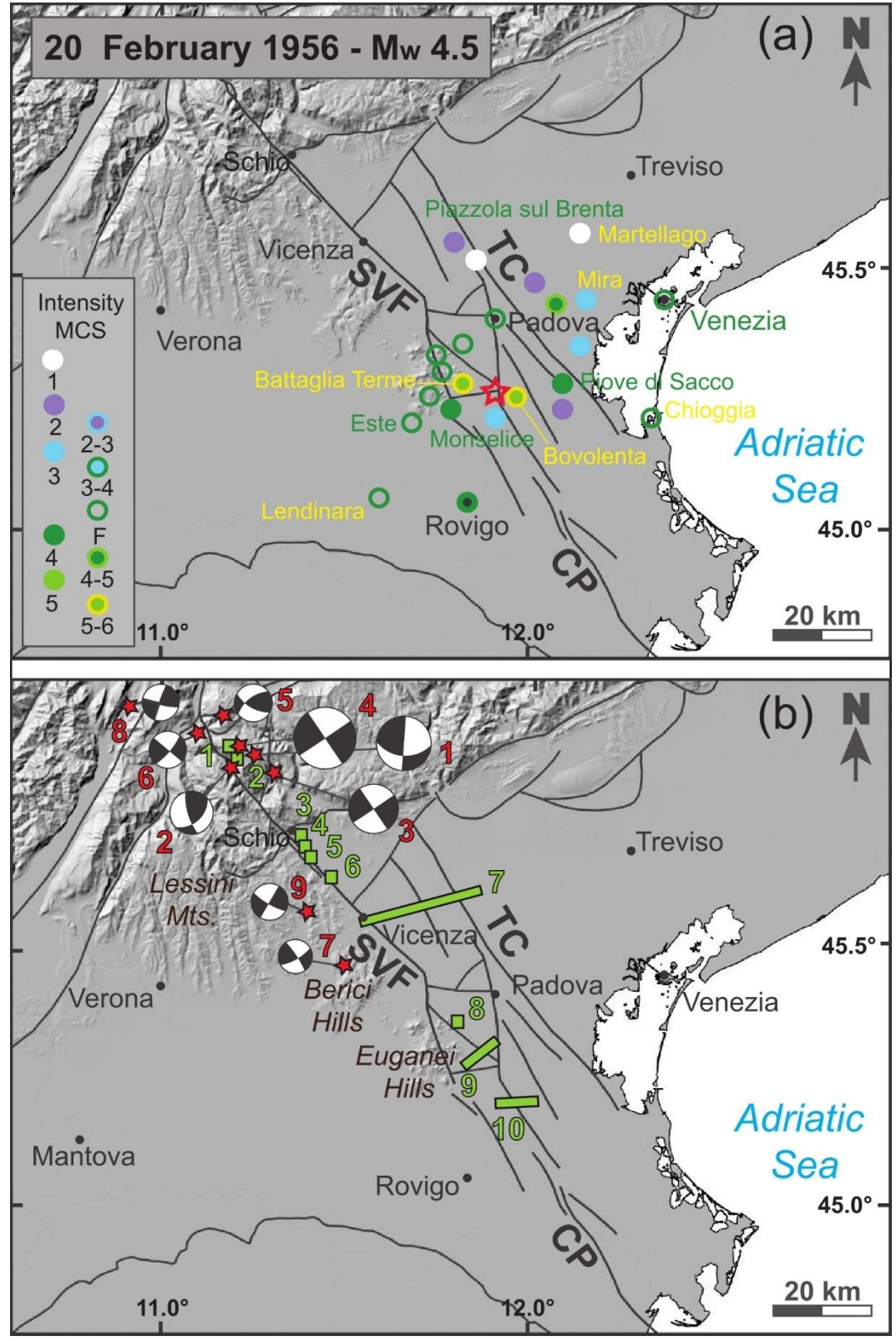

**Figure 4**. (a) Macroseismic dataset of the 20 February 1956, $M_w$ 4.5, earthquake with intensities shown in MCS scale. The selected reference macroseismic study for this event in the CPTI15 catalogue (Rovida et al., 2021), is the work by Caracciolo et al. (2009). The instrumental epicenter is shown with a red star, coincides with the macroseismic one, and falls near the SVFS. Due to its location, one of the mapped faults belonging to the SVFS may be the source of this earthquake (see text and Table 3). (b) Map showing the largest instrumental earthquakes of the study area with red stars (see Table 4). Focal mechanisms from EMMA Database and other sources (Vannucci and Gasperini, 2004; Slejko and Rebez, 1988; Slejko et al., 1989; Pondrelli et al., 2006; Viganò et al., 2008). Earthquake relocations from Viganò et al., 2015. The locations of the geological evidence are shown in green (see Table 2, Fig. 2, and text for details). These focal mechanisms are compatible with a right-lateral strike-

slip reactivation of the faults belonging to the SVFS. Digital elevation model of central map is EU-DEM v1.1. Available online: https://land.copernicus.eu/imagery-in-situ/eu-dem (accessed on 20 June 2021).

Instrumental catalogues show that the SVFS area is affected by low seismic moment release, with most events located in its northern mountainous portion, north of Schio (Fig. 4b and Table 4), and have hypocentral depths ranging between 2 and 19 km (Viganò et al., 2008; Moratto et al., 2017). Their moment magnitude ranges between 3.3 and 4.9. The 13 September 1989, $M_w$ 4.9, earthquake (ID 4 in Table 4 and Fig. 4b) is the largest instrumental event that affected the study area, and we consider it as representative of the local kinematic style of faulting of the SVFS. The macroseismic effects of this event are documented

by 779 intensity points spread over a wide area covering the Southern Alps and the Po and Veneto plains (Rovida et al., 2021). The epicentral intensity of the 1989 shock is VI-VII MCS, and the area of greatest damage lies in the Giudicarie-Lessini region, north of the locality of Posina. The focal mechanism of the 1989 event is strike-slip (Eva and Pastore, 1993; Pondrelli et al., 2006; Viganò et al., 2008), and the direction of its maximum horizontal compressive stress is compatible with right-lateral strike-slip activation of a NW-SE trending fault belonging to the northern portion of the SVFS (Viganò et al., 2008; Vannoli

et al., 2015; Restivo et al., 2016).

All available focal mechanisms of the study area consistently show right-lateral strike-slip over the northern portion of the SVFS (Table 4 and Fig. 4b; Vannucci and Gasperini, 2004; Viganò et al., 2008).

| ID | Date | Time *hh:mm:ss* | Lat *deg* | Lon *deg* | Depth *km* | $M_w$ | Strike *deg* | Dip *deg* | Rake *deg* | FM Ref |
|----|------|------|------|------|------|------|------|------|------|------|
| 1 | 22 Jun 1968 | 12:21:37 | 45.880 | 11.280 | 9 | 4.7 | 94;3.5 | 41;90 | 0;131 | Sle89 |
| 2 | 22 Jun 1968 | 12:27:49 | 45.840 | 11.210 | 19 | 4.1 | 48;164 | 40;70 | 148;55 | Sle89 |
| 3 | 21 Jul 1983 | 13:31:22 | 45.846 | 11.324 | 12 | 4.3 | 252;82 | 50;40 | 84;98 | SR88 |
| 4 | 13 Sep 1989 | 21:54:02 | 45.882 | 11.264 | 9 | 4.9 | 146;56 | 90;89 | -179;0 | Pon06 |
| 5 | 24 Oct 1994 | 23:22:48 | 45.938 | 11.188 | 8 | 3.5 | 120;233 | 60;56 | 140;37 | Vig08 |
| 6 | 26 Apr 1999 | 02:53:46 | 45.916 | 11.096 | 2 | 3.4 | 40;310 | 75;90 | 0;165 | Vig08 |
| 7 | 28 Oct 1999 | 10:16:14 | 45.453 | 11.451 | 8 | 3.3 | 60;327 | 70;81 | 10;160 | Vig08 |
| 8 | 16 Jun 2000 | 11:57:22 | 45.989 | 10.927 | 5 | 3.4 | 105;198 | 75;80 | 170;15 | Vig08 |
| 9 | 18 May 2005 | 21:41:09 | 45.569 | 11.385 | 4 | 3.4 | 30;120 | 90;75 | -15;-180 | Vig08 |

**Table 4**: Post-WWSSN (World-Wide Standardized Seismograph Network) located near the trace of the SVFS and for which were calculated focal mechanisms. FM Ref, reference for the focal mechanisms: Slejko and Rebez, 1988 (SR88); Slejko et al., 1989 (Sle89); Pondrelli et al., 2006 (Pon06); Viganò et al., 2008 (Vig08). Location of earthquakes ID 1 and 2 from Sandron et al., 2014; earthquakes ID 3 and 4 from the OGS Bulletin (http://www.crs.inogs.it/bollettino/RSFVG/). The locations of these four earthquakes are also reported in the CPTI15 Catalogue (Rovida et al., 2021). ID 5 to 9, earthquakes relocated by Viganò et al., 2015 having a magnitude type $M_d$.

**5 Discussion**

The SVFS has been active with different tectonic phases and different kinematics at least from the Mesozoic (Table 1). Its most prominent segment, the SVF s.s., is an inherited and well-developed morphotectonic element with a significant imprint in the landscape, reactivated in the current stress regime as testified by the moderate seismicity along its northern portion (Table 4), and the travertine recent deposits in its central section (ID 8 in Fig. 2 and Table 2; Pola et al., 2014a). Further

possible evidence of the activity and of the seismogenic potential of the SVFS, come from the historical seismicity registered near Padova and Vicenza (Figs. 3 and 4a; Table 3).

Looking at the geological and geophysical evidence south of Posina (IDs 3 to 9 of Fig. 2 and Table 2) some consistent conclusions may be drawn about the geodynamic role, the recent activity and style of faulting of the SVFS. Pleistocene differential vertical movements across the fault system occurred close to Schio, as shown by the differential uplift of the western block (footwall) of the SVF with respect to the plain (Pellegrini, 1988; ID 3). Accordingly, in the early Pliocene marine conditions were present in the eastern block of the fault, while continental conditions characterised the western block (Zanferrari et al. 1982). The same evidence is shown by the attitude of Miocene sediments along the northern part of the SVF between Schio and Vicenza, where the deposits dip 50° towards northeast, being the limb of a drag fold in the footwall of an extensional fault (IDs 4 to 6). The same structure is recognizable in some seismic sections sub-orthogonal to the SVF (Pola et al., 2014b; ID 7).

Therefore, several lines of evidence testify the extensional component of the fault, especially south of Schio, while the strike-slip component is not well constrained. However, geodynamic considerations point to a role of the SVFS complementary to the NW-striking Dinaric system (e.g. Predjama, Idrija and Ravne faults), acting presently as dextral strike-slip faults and accommodating to the east the indentation of Adria against the Alpine chain together with the transpressional faults of the external Dinarides (e.g. Moulin et al., 2016; Heberer et al., 2017; Atanackov et al., 2021). If so, the SVFS, having the same NW-SE strike of the Dinaric system, but delimiting the Adriatic indenter to the west, on the whole should have a sinistral component of motion. This is what was commonly suggested by a number of scholars studying the geology of the northeastern part of Italy (Table 1).

However, the few instrumental seismological data recorded near the SVF northwesternmost part (north of Posina), point to a dextral strike-slip activity (Pondrelli et al., 2006, 2020; Viganò et al., 2008, 2015). In addition, the northwestern part of the SVF, close to the Borcola Pass, subvertical conjugate fault planes crop out in two quarries exploiting Upper Triassic dolomitic marbles produced by intrusion of basaltic dykes. The wide slickensides show sub-horizontal corrugations produced by nearly pure sinistral strike-slip movements (IDs 1 and 2 of Fig. 2 and Table 2). The analysis of the kinematic indicators suggests a sinistral activity of these N-S faults, which has permitted to infer a dextral kinematics for the SVF (Fondriest et al., 2012).

Therefore, this review of published papers presents a prominent problem about interpretation of the recent strike-slip sense of motion of the SVF, which is described as sinistral (in most cases), but also as dextral (Tables 1 and 2). A model to solve the apparently conflicting data must be found considering the complexity and the 3D structure of the SVFS, which is more than a straight line as usually drawn in the structural maps.

If we look at the plan view geometry of the SVFS in the chain sector (north of Schio), we observe a junction with the Gamonda fault (GF) close to Posina, ca 10 km northwest of Schio (Fig. 5; Sedea and Di Lallo, 1986; Zampieri et al., 2003; Massironi et al., 2006; Zampieri and Massironi, 2007). The NW-trending SVF and the NNE-trending GF depicts a Y-type triple junction, with transport direction at a high-angle to the branch line, which is sub-vertical. This classification derives from the very detailed analysis of the shear zone and brittle faults junctions, also called zipper junctions, performed by Passchier and Platt (2017; see inset in Fig. 5). A wide variety of possible configurations for merging faults is possible. In our case, the plan view geometry of the junction falls within the "sinistral opening zipper" model, where the SVF south of Posina can have a sinistral strike-slip movement likewise the GF, while the SVF north of Posina must experience a dextral movement (Fig. 5). The space problem produced by the intersection of simultaneously active faults with opposing slip sense can be solved by splitting (or unzippering) the merged faults (Platt and Passchier, 2016). Not being supported by the enlightening paper of Passchier and Platt (2017), an earlier solution to the triple junction of the SVFS was presented by Zampieri et al. (2003), where the SVFS branch north of Posina was depicted to slip only moderately (i.e. less than the segment south of Posina) with sinistral activity. In the work of Zampieri et al. (2003) the area north of the Posina Triple Junction, bordered by the SVF and the N-S trending

faults (i.e. GF-TF-MeF-CF in Fig. 5) has been interpreted as a pop-up rising within a restraining bend of the SVF (Fig. 5). Close to the Posina Triple Junction, this wedge presents contractional structures such as reverse faults and folds (Zampieri et al., 2003), in line with the patterns expected by the Passchier and Platt (2017) model.

According to these models, it is possible that the portion of the SVFS north of the triple junction (PTJ in Fig. 5) has experienced repeated opposite strike-slip movements over time. The emplacement of N-trending basaltic dykes north of PTJ, in the damage zone of the SVF suggests dextral transtension (Zampieri et al., 2003; Fondriest et al., 2012) during the Paleogene, when the area was deformed by nearly E-W extension (Zampieri, 1995). During the initial stages of the Neogene contraction, the SVF may have slipped by sinistral movements, partly inverted by subsequent dextral movements, when the wedge structure north

of the triple junction opened the zipper of the merging faults.

As a matter of fact, the opening zipper model shows right-lateral motion north of Posina and left-lateral motion to the south as well as along the N-S segments (Zampieri and Massironi, 2007), in agreement with the available focal mechanisms of the largest earthquakes and to the kinematic model presented by Serpelloni et al. (2016, see Fig. 8). Furthermore, the left-lateral kinematics of the SVFS up to the Posina triple junction at the regional scale is also in agreement with the geodynamic model

of the Adria plate indentation (see Fig. 7), while the zipper structure represents a local feature due to the back-stop of the Giudicarie thrust front (see also Verwater et al., 2021 for an alternative interpretation).

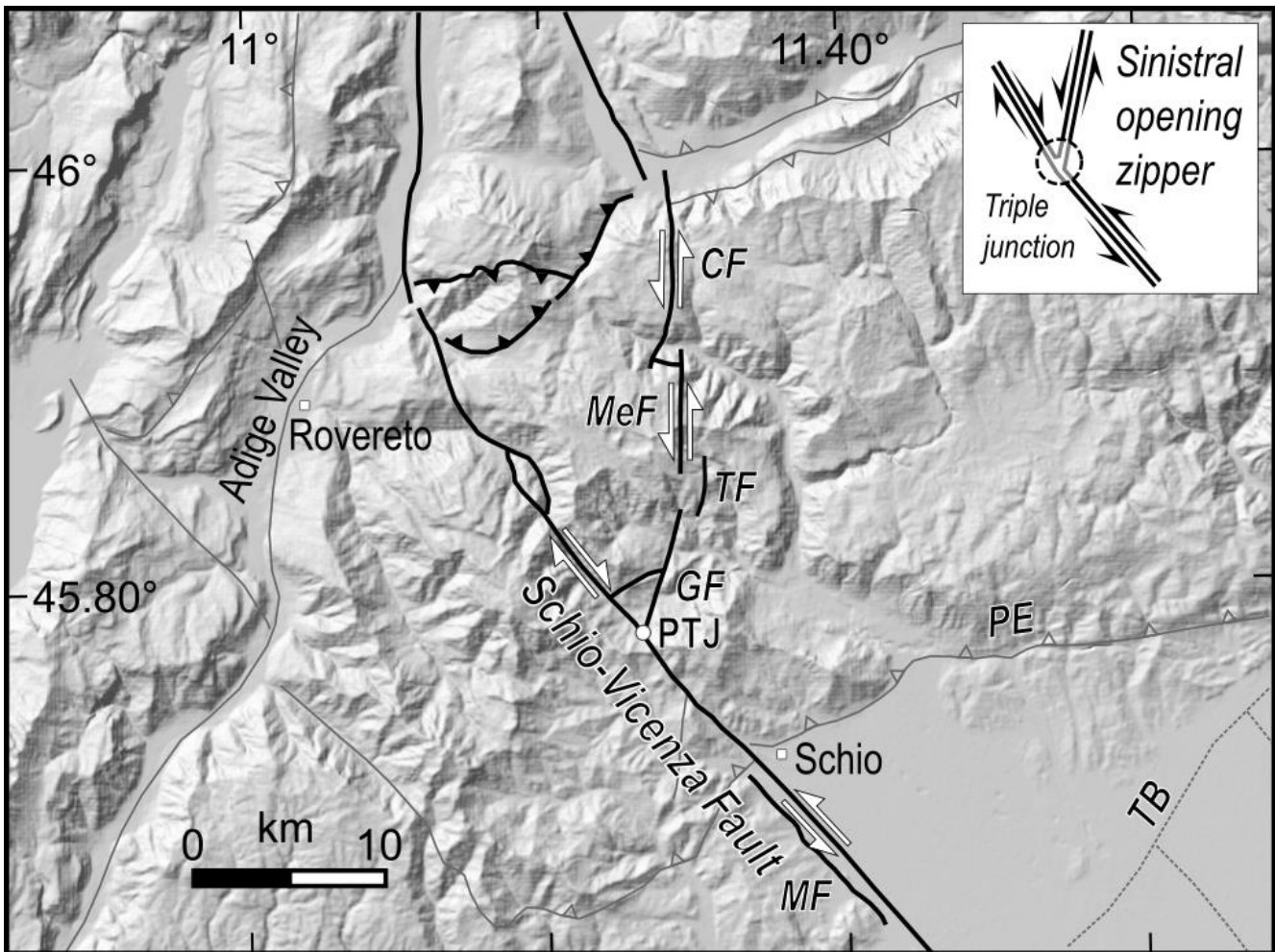

**Figure 5**. Intersection (PTJ: Posina Triple Junction) of the Schio-Vicenza fault and a set of inherited linked faults (GF:
Gamonda fault, TF: Tormeno fault, MeF: Melegnon fault, CF: Carotte fault) on the western side of Eastern Southern Alps. The space problem produced by the intersection of simultaneously active faults with opposing slip sense can be solved by splitting (or unzippering) the merged fault (inspired by Zampieri et al., 2003; Zampieri and Massironi, 2007; inset of the right corner from Passchier and Platt, 2017). PE: Pedemontana thrust; TB: Thiene-Bassano buried fault. The kinematics refers to

the last Neogene activity, possibly extending until the Present. Digital elevation model of central map is EU-DEM v1.1.
Available online: https://land.copernicus.eu/imagery-in-situ/eu-dem (accessed on 20 June 2021).

The study of the interaction of strike-slip dominated structures at high angle with thrust belt fronts has been carried out by Fedorik et al. (2019) by means of analogue model experiments using quartz sand as analogue material of the brittle upper crust. The models were run simulating different tectonic regimes, from pure strike-slip, to transpressional and transtensional ones, and studied how a trascurrent fault system evolved and interacted with pre-existing and newly formed thrust fronts. The pure strike-slip regime was simulated using a moving plate shifting parallel to the strike of the main trascurrent fault, while transpression and transtension were simulated making the moving plate shifting in directions forming various angles, from 10° to 30°, with respect to the orientation of the trascurrent system (for further details see the paper by Fedorik et al., 2019). The 30° transtensional model (Fedorik et al., 2019, Fig. 15) produces some Riedel shear faulting and is dominated by two major steep strike-slip faults having also a normal dip-slip component. The last stage of deformation also shows the linkage between the strike-slip dominated fault zone and the forethrust by the development of an oblique thrust (Fig. 6). This transtensional setting resembles very closely to the real structure composed by the SVF and the Eastern Southern Alps thrust front. In fact, the northern Adria plate is pushing with a NNW direction against the Alpine belt, with a small angle with respect to the NW-trending SVF (see Figs. 1 and 7), and the kinematic modeling of the GPS data shows transtension on the SVF (Serpelloni et al., 2016) in agreement with the dip-slip displacement shown by the available seismic sections crossing the SVFS (Fig. 2). Besides, the kinematics and the relative position of the strike-slip faults with respect to the chain front are similar in both real geology and the models. The only difference of the analogue modelling with respect to the real structure is the timing of the deformation phases. In the models the thrust belt development predates the strike-slip activation in the foreland, while in the study area the development of the SVF predates that of the frontal thrusts of the chain. However, the SVF activity continues after the thrust formation and the interaction between the two structures is very similar to the model. In fact, the buried TB fault, which connects obliquely the PE thrust with the SVF (Figs. 6a and 6b) was reproduced by the models and corresponds to the new thrust front of Fig. 6c. The similarity of the analogue modelling of Fedorik et al. (2019) with the real setting of the study area strongly supports the interpretation of the SVFS south of Schio as a sinistral strike-slip fault.

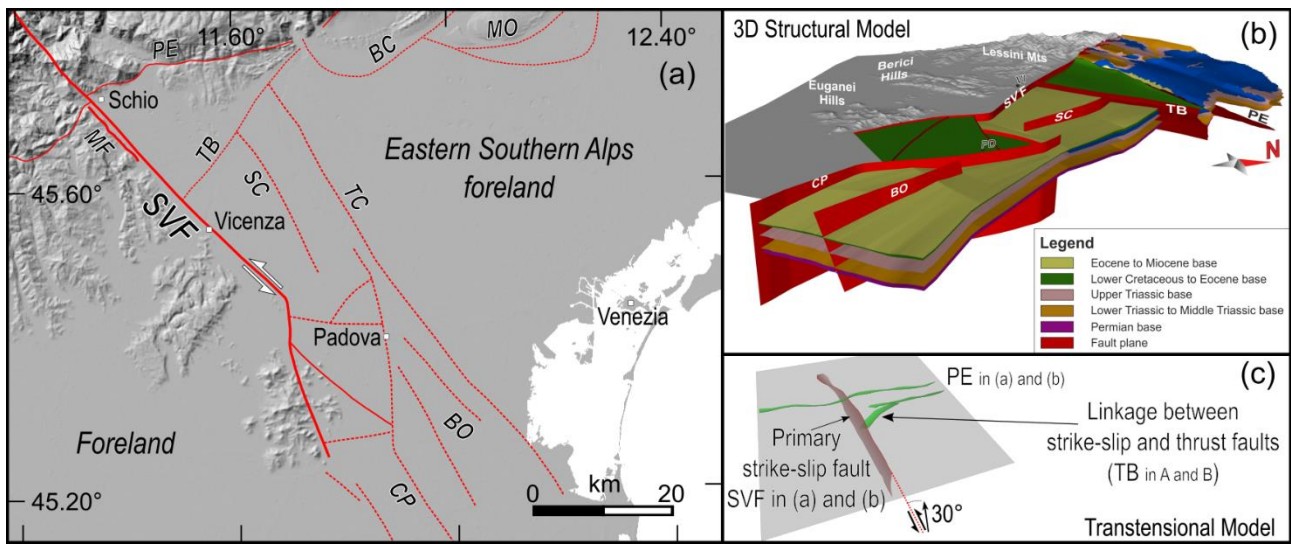

**Figure 6**. SVFS in map, in 3D view and its analogue model. (a) structural sketch of the SVFS (BC: Bassano-Cornuda thrust; BO: Bovolenta fault; CP: Conselve-Pomposa fault; MF: Malo fault; MO: Montello thrust; PE: Pedemontana thrust; SC: Sandrigo-Camisano fault; SVF: Schio-Vicenza fault; TB: Thiene-Bassano thrust; TC: Travettore-Codevigo fault); (b) 3D reconstruction of the SVFS, adapted and modified from Torresan et al. 2020; (c) sketch derived from the 30°-transtensional model showing the interaction between a strike-slip dominated fault zone and thrust belt structures (from Fedorik et al. 2019). Notice that this model predicts the development of an oblique thrust fault connecting the frontal thrust of the chain and the transverse structure, corresponding to the TB thrust of panels (a) and (b). Digital elevation model of map (a) is EU-DEM v1.1. Available online: https://land.copernicus.eu/imagery-in-situ/eu-dem (accessed on 20 June 2021).

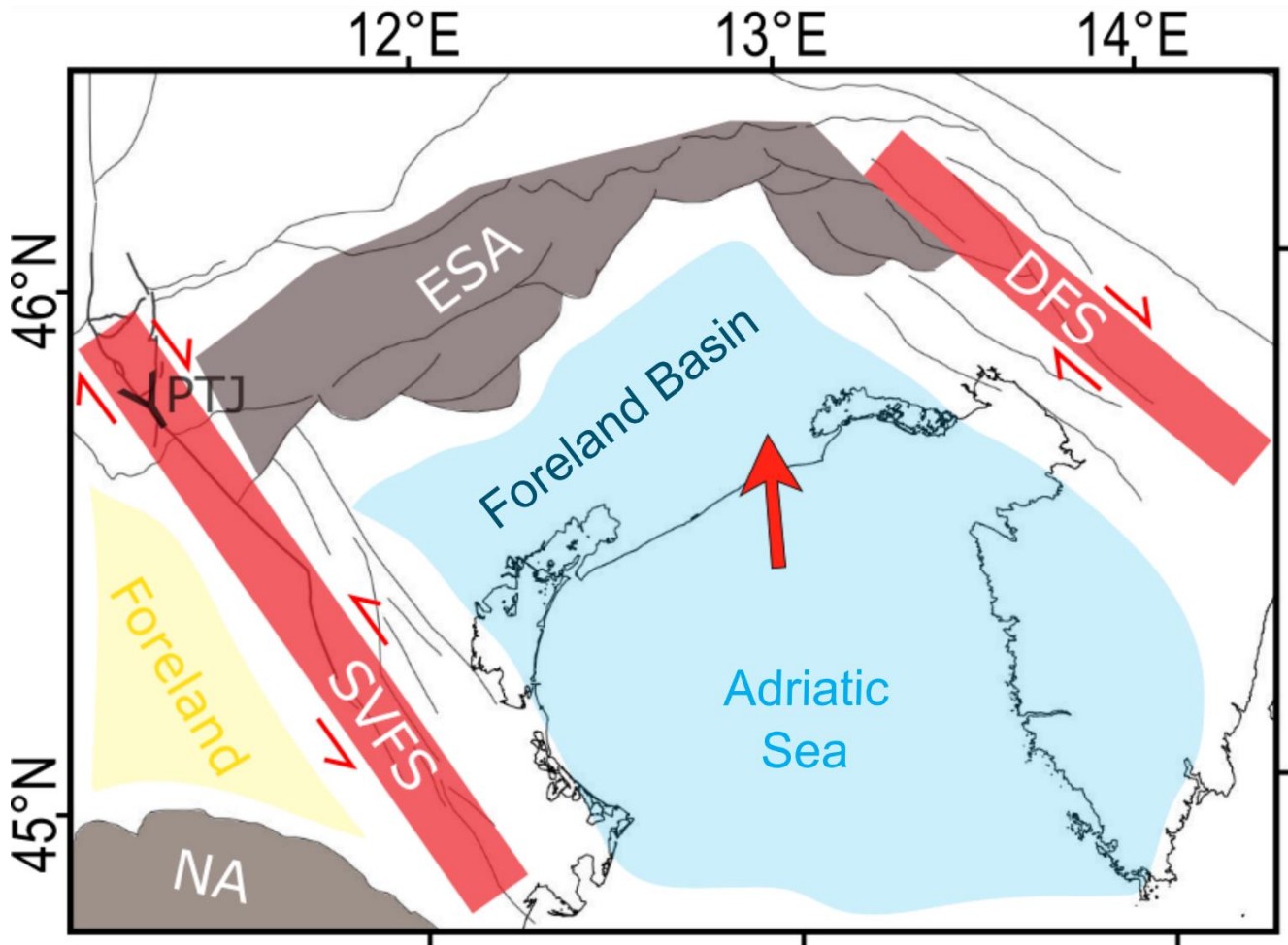

**Figure 7**. Schematic sketch of the North Adriatic (Dolomites) indenter model bounded by the SVFS to southwest and the dextral transpressional Dinaric Fault System (DFS) to northeast. Notice the sinistral transtensional movement of the SVFS south of Posina Triple Junction (PTJ) according to the indenter model, and the dextral strike-slip movement north of PTJ, according to the zipper model (shown with a Y). ESA: Eastern Southern Alps, corresponding to the northern shortened Adria plate margin; NA: Northern Apennines outermost front. The red arrow shows the average direction of the relative motion of Adria with respect to fixed Eurasia (Serpelloni et al., 2016; Devoti et al., 2017).

The meaning of the SVFS in the framework of the regional tectonics and its long-lived history is shared with other fault systems found in the thrust belt and foreland of the central and southern Apennines and Sicily. These shear zones are inherited structures reactivated in the current stress regime driven by the plate convergence, and can host the sources of moderate to large earthquakes (e.g. Tavarnelli et al., 2001; Butler et al., 2006; Di Bucci et al., 2010). The most important of them are: (a) the Mattinata-Gondola fault system in the Gargano area, a segment of which located along its western prolongation generated the $M_w$ 5.7, 2002 Molise seismic sequence (Valensise et al., 2004; Di Luccio et al., 2005), and is probably the source of the $M_w$ 6.7, 1627 earthquake (DISS Working Group, 2018); (b) the E-W striking fault system associated with the $M_w$ 5.8, 1990 Potenza earthquake (Di Luccio et al., 2005; Boncio et al., 2007; DISS Working Group, 2018); (c) the Scicli-Ragusa fault system (Di Bucci et al., 2010); and (d) the Olevano-Antrodoco fault (Tavarnelli et al., 2001; Butler et al., 2006; Bonini et al., 2019; Vannoli et al., 2021). All these fault systems, during their long history, have seen different tectonic phases often characterised by opposite sense of shear (e.g. first right-lateral strike-slip, then left-lateral strike-slip or vice-versa). As a consequence, as seen during the 1990 and 2002 seismic sequences, these inherited fault systems are often characterised by low stress drops and, consequently, small coseismic slip (Calderoni et al., 2010). In addition, these large long-lived shear zones are responsible in Italy for the largest seismic release outside the better known active thrust and extensional belts, and play a primary role in the seismotectonics of the central Mediterranean region (DISS Working Group, 2018).

The Adriatic plate indenter has been considered as delimited by the Giudicarie fault system to the west, the Pustertal-Gailtal fault to the north (Ratschbacher et al., 1991; Reiter et al., 2018; Rosenberg et al., 2004; Rosenberg et al., 2007) and the Dinaric fault system to the east (Mantovani et al., 2006; Massironi et al., 2006; Heberer et al., 2017; Brancolini et al., 2019). In this context, the SVFS represents an intraplate structure crossing the Mesozoic Trento platform. However, since the late Messinian the Adriatic block decoupled from its nearly stationary northwestern (Padanian) protuberance and the western indenter margin widened incorporating the SVFS, while leaving outside the contractional deformation the Lessini-Berici-Euganei foreland block (Mantovani et al., 2006; Massironi et al., 2006;), which represents an anomalous tectonic feature separating the Western Southern Alps from the ESA and the related forelands (Laubscher, 1996). In fact, tectonic balancing of cross sections parallel to the local NNW-SSE shortening direction has suggested that the ESA comprise two kinematic domains that accommodated different amount of shortening during overlapping times (Verwater et al., 2021). These two domains are separated by the Trento-Cles – Schio-Vicenza fault systems, which offsets the ESA front in the south, while merging with the Northern Giudicarie fault to the north (Fig. 1). The east-west lateral variation of shortening across the Giudicarie Belt indicates internal deformation and lateral variation in strength of the Adriatic indenter related to Permian - Mesozoic tectonic structures and paleogeographic domains (Verwater et al., 2021).

## 6 Conclusions

In the northern part of the Adria plate the transverse structure of the SVFS has been active with different tectonic phases and different kinematics at least from the Mesozoic. Nowadays, its westernmost segment (SVF s.s., i.e. the segment from the Adige Valley to the Euganei Hills) is an inherited and well-developed structural element with a significant imprint in the landscape, reactivated in the current stress regime. If the long-term dip-slip component of faulting is evident on the SVF and on other buried segments of the fault system, on the contrary, the horizontal component of the movement is not well constrained although it could have accommodated a total of a few kilometers of displacement with sinistral motion (Table 1).

Unfortunately, apart from the moderate instrumental seismicity near its northern end, the moderate historical seismicity near its central sector, and the geological evidence of recent deformation of a travertine mound close to the Euganei Hills, there is little evidence to constrain the recent activity of the SVFS, and even its role in the geodynamic framework of the Southern Alps is still a matter of debate. Although its kinematics is still largely unknown, we observe that it interrupts the continuity of the Southern Alps thrust fronts in the Veneto sector, and controls the forward propagation of the thrusts, suggesting that it played a key passive role in controlling the geometry of the active fault systems and the current distribution of seismic release. As a matter of fact, the true kinematics of this fault system is ambiguous, since the instrumental data along the northernmost part (to the north of Posina) of the SVFS point to a dextral strike-slip activity. The most important event is the 13 September 1989, $M_w$ 4.9 earthquake, located north of Schio. While further evidence on the kinematics of the SVFS shall arise solely from the results of further research in the area, the apparently conflicting data can be reconciled in the sinistral opening "zipper" model, where intersecting pairs of simultaneously active faults with different sense of shear merge into a single fault via a zippered section. The junction of the three branches would be located at Posina, about 10 km northeast of Schio. The comparison with available analogue models supports a sinistral strike-slip kinematics for the main portion of the SVF from the plain to the Posina triple junction. The debate on the strike-slip kinematics of the SVF has been tentatively solved and seems to work unless new data will threaten the model presented here.

The SVF does not appear to have generated earthquakes larger than magnitude 5.0. Moderate instrumental seismicity is clustered in the northern portion of this structure, and historical catalogues show a scattered distribution of low-magnitude earthquakes along its central and southern portions. However, the set of geomorphological, structural and geodynamic features suggest that individual segments of the SVFS may host significant (M 5.5+) earthquakes. As discussed in chapter 4, the

causative fault of the largest known earthquake of the Po Plain, the 1117 $M_w$ 6.5 Veronese earthquake, is believed to be a long-lived, inherited fault cutting the foreland.

Therefore, although the earthquake potential of the SVFS is still uncertain, and need to be fully characterised, these long-lived
transverse structures cutting through the foreland, such as the SVFS, may represent a substantial, and still poorly recognised source of seismic hazard for the area.

**Author contribution**

All the authors shared the conceptualization of the manuscript and its preparation.

**Competing interests**

The authors declare that they have no conflict of interest.

**Acknowledgements**

We thank Filippo Torresan and Giovanni Toscani for discussion about their models presented in our Figure 6, the 3D model of the SVFS and the analogue modeling of a transverse structure, respectively. DZ acknowledge *Università degli Studi di Padova* for DOR research fundings. We also thank Enrico Tavarnelli, Maria Eliana Poli and Hugo Ortner for their thorough
review that greatly improved the manuscript.

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
