# Peer review of "Geodynamic and seismotectonic model of a long-lived transverse structure: The Schio-Vicenza Fault System (NE Italy)"

_Solid Earth, 2021_

## Referee Comment (RC1)

Review report by Enrico Tavarnelli for manuscript n. se-2021-29 by Dario Zampieri, Paola Vannoli and Pierfrancesco Burrato titled: "*Geodynamic and seismotectonic model of a long-lived transverse structure: The Schio-Vicenza Fault System (NE Italy)*", submitted to Solid Earth.

This paper provides a new and original documentation of the structural history and seismotectonic evolution of a long-lived transverse lineament, the Schio-Vicenza Fault System, that dissects the thrust front of the Southern Alps in the Veneto sector of the Adria foreland microplate in NE Italy. This is achieved through a detailed and genuinely multidisciplinary approach, that integrates field mapping, stratigraphic investigation, structural analysis, seismic profile interpretation, current and historical seismicity, coupled with a wealth of data from a wide literature in the region. The topic dealt with in the study is of great interest to anyone that has an interest in understanding the evolution of the Southern Alpine system and in constraining the seismotectonic potential of the area. Moreover, the study illustrates an example of applicability of universal concepts of fault reactivation and structural inheritance under a 3D view, with emphasis on the interaction of strike-slip systems at triple junctions. These topics have generated a lively debate, and the submitted manuscript sheds new light in this direction, providing a very well-documented case. The Authors' interpretations are consistent with the data presented, and the original "zipper model" proposed to account for strike-slip reversals is very convincing.

The manuscript is well written and well organised, with English and presentation forms that are overall very good. The illustrations and tables are all clear, legible and very much informative (but see my separate comment to Fig. 2). The quality of the contribution, in all its parts, is overall high. Good credit is given to the existing literature, both methodological and regional. However, I believe that the manuscript would benefit from a slight extension of the reference list, with citation of a few papers that are listed separately in this review report. Unfortunately, part of the suggested missing references happen to arise from my own research, and in general I am quite reluctant to self-advertise my work amongst colleagues. But the submitted manuscript refers to topics where my collaborators and I have long worked and published; thus I believe that a slight extension of the reference list with inclusion of the mentioned contributions would be highly beneficial for the reader with field examples in thrust belts that laterally flank the South Alpine chain.

I found this an extremely stimulating contribution and believe that it will make a very interesting title for a genuinely international and multidisciplinary audience. It is my opinion that the manuscript may be accepted for publication almost as it stands, with only the incorporation of a few sentences (with related references listed below), and the insertion of minor alterations to the text for the sake of an improved legibility. Therefore, I recommend without reservations that this manuscript is accepted for publication on Solid Earth only pending on minor suggested revisions, that are listed separately.

I require no anonimity and wish that all my comments are forwarded to the Authors. I hope that my review is received as a constructive and supportive indication, that may assist the Authors to achieve an even more suitable and documentally supported paper, and the Editor in formulating a final, positive decision in the interest of Solid Earth and of its wide, international readership.

Siena, Italy, 24 May 2021

Sincerely,

Enrico Tavarnelli

A) LIST OF SUGGESTED ALTERATIONS:

Note – the parts outlined in red are suggested to be removed; the parts outlined in green are suggested to be inserted/incorporated; the parts outlined in yellow require the Authors' attention during their revision.

Page 2, Line 39: "… Carrillo et al., 2020), in the Northern-Central Apennines (Tavarnelli et al., 2001; Butler et al., 2006; Peacock et al., 2017), in the Southern Apennines foreland… ". NOTE: the reference to these papers are listed separately in the forthcoming B) section of my review report.

Page 5, Line 100: "… in the Veneto Plain (Fig. 1),  was drawn…

Page 7, Line 187: "…  Southern Alps,  whose compressional deformation…".

Page 7, Line 188: "… Eastern Southern Alps,  whose deformation is still active."

Page 10, Line 282: "… two of which  are presented in Fig. 2 (IDs 6 and 8)…"

Page 10, Line 282 – and also Page 9, Fig. 2 – Fig. 2, IDs 6, referred to in the text, is not indicated in the Figure. The following is stated in  the Figure caption: "The IDs 4 and 6 can be found in the sheets 36 Schio (Braga et al., 1968) and 49 Verona (Bosellini et al., 1968) of the 1: 100,000 scale Geological Map of Italy. See also Table 2", but this indication is not straightforwards for the reader (al least, for this reader). It would be useful if the Authors could attempt at providing, or sketching, or summarizing, the missing information within the manuscript. Should this not be possible, the Authors should clearly indicate the missing documentation directly in the text, as they have already done in the Figure caption.

Page 10, Line 303: "… are  S-directed blind thrusts…". The use of vergence is inappropriate here, since vergence is a property of folding, not faulting. The meaning of the sentence is still clear, but I would suggest to avoid the conceptually wrong term " " and to replace it with the more correct "S-directed".

Page 12, Line 351: "… from the SVFL, and  don't belong to it."

Page 14, Line 379: "… with most  events…".

Page 16, Line 431: "… branchline…". In general this is spelt with two separate words in structural accounts: branch line. I would suggest that this notation is used in the manuscript.

Page 16, Line 438: "… Zampieri et al. (2003), …". I would change this sentence as follows: "… Zampieri et al. (2003), where the SVFS branch…".

Page 17, Line 461: "The study of…  been conducted by Fedorik et al. (2019) by means of…". I would rephrase this sentence as follows: "The study of… has been carried out by Fedorik et al. (2019) by means of…".

Page 17, Line 476: "new thrust front of  Fig. 6c.".

Page 18, Line 497: "… moderate to large earthquakes (e.g. Tavarnelli et al., 2001; Butler et al., 2006; Di Bucci et al., 2010).

Page 19, Line 503. I would add another important cross-lineament in the northern-central Apennines, (as a separate case.d) whose history was, similarly, characterized by a strike-slip reversal reactivation, from sinistral to dextral: this reversal occurred along the Ancona-Anzio Line (or Olevano Antrodoco Fault), as

described by ==Tavarnelli et al., 2001== and by ==Butler et al., 2006==. Indeed, the following statement, already present in the text ("All these fault systems, during their long history, have seen different tectonic phases often characterised by opposite sense of shear (e.g. first right-lateral strike slip, then left-lateral strike-slip, ==or vice-versa=="), perfectly apply to the history described by ==Tavarnelli et al. 2001== (see their Fig. 11a and 11b) and by ==Butler et al. 2006== (see their Fig. 10a and 10b).

Page 19, Line 515: "…  ==accommodated==… "

B) LIST OF SUGGESTED REFERENCES TO BE INCORPORATED AND ACKNOWLEDGED:

Tavarnelli E., Decandia F.A., Renda P., Tramutoli M., Gueguen E. & Alberti M. (2001) - *Repeated reactivation in the Apennine-Maghrebide system, Italy: a possible example of fault-zone weakening?* Geological Society of London Special Publication 186, "The Nature and Tectonic Significance of Fault Zone Weakening" (Holdsworth, R.E., Strachan, R.A., Maglouglin, J.F. & Knipe, R.J. , Eds.), 273-286.

Butler, R.W.H., Tavarnelli, E. & Grasso, M. (2006) – *Structural inheritance in mountain belts: an Alpine-Apennine perspective.*. Journal of Structural Geology, 28, 1891-1892.

Peacock, D.C.P., Tavarnelli, E. & Anderson M. W. (2017) - *Interplay between stress permutations and overpressure to cause strike-slip faulting during tectonic inversion*. Terra Nova, 29, 61-70. doi: 10.1111/ter.12249

Enrico Tavarnelli

---

## Referee Comment (RC2)

65

[referee-annotated manuscript omitted]

---

## Referee Comment (RC3)

[referee-annotated manuscript omitted]

---

## Author Response (AR1)

We would like to thank and express our appreciation for the positive reviews of the three Reviewers and acknowledge the insightful comments and stimulating questions that helped to improve the manuscript.

**The Reviewer #1 raised the following main points.**

1. **Extension of the reference list and discussing other Italian examples of inherited, reactivated fault systems.**

We accepted the suggestion given by the Reviewer, that included also extending the discussion to the Olevano-Antrodoco Fault System as another natural example of long-lived, inherited fault system reactivated in the present-day active stress field.

**There are three main points raised by the Reviewer #2.**

1. **The first one is related to the completeness of the historical seismic catalogue and has implications for our hypothesis that small magnitude earthquakes occurred along the SVFS in the past years.**

There are two kinds of completeness that can be discussed: one regarding the earthquake list included in the catalogue, and another one regarding the completeness of the macroseismic information related to the single earthquake. In both cases, the completeness of information is related to the magnitude and age of the event considered, and generally for lower magnitude earthquakes the threshold of completeness is closer to the present day than for higher magnitude (i.e. we have a longer completeness for the higher magnitude events). We think that further discussing this topic is beyond the scope of the paper (for further reading, see for example: Stucchi, M., Albini, P., Mirto, C., and Rebez, A. (2004), Assessing the completeness of Italian historical earthquake data, Ann. Geophys. 47, 2–3, 659–673, doi:10.4401/ag-3330).

In any case, the hypothesis we discuss in the text, a number of small magnitude earthquakes felt only in Padova or Vicenza and occurred along the SVFS, derives from the following observations:

1) the earthquakes listed with only one macroseismic data point are real earthquakes, otherwise they would not have been listed in the Italian historical catalogue (the authors of the catalogue found historical documents describing the events);

2) they may be low magnitude events because the two cities are very close to each other and high magnitude events could have been felt in both localities;

3) it is true that the macroseismic dataset for these events may be incomplete (due for example to lost historical sources or never written sources), but this is more probable to happen for low magnitude events producing small macroseismic effects, and also for the same reason they have been registered only in the main city of the epicentral area and consequently their true epicentral location is not really the city itself but they may be located somewhere nearby;

4) being very probably low energy local events occurred close to the main cities, our hypothesis is that they may have been generated by the SVFS (the only mapped active fault system of the area);

5) finally, these historical events may have been similar to the 20 February 1956 earthquake.

2. **The second main point raised is related to the focal mechanisms of the small magnitude earthquakes shown in Figure 4, and if they can be used to infer the kinematics of the active faults present in the area.**

Most of the earthquakes shown with the focal mechanisms are small magnitude events most probably not located on the main faults forming the opening zipper system. So they cannot be used to infer directly the

kinematics of this fault system. However, their focal mechanisms are in agreement with the regional stress regime that is the engine responsible for the kinematics of the different faults.

The same is true for the earthquakes #7 and 9, whose causative faults according to their magnitude are in the order of few hundreds of meters long and may be secondary structures located in the footwall block of the SVF.

Of all the events shown in Figure 4, the most reliable is the #4, that due to its magnitude is the better located and is compatible with the right-lateral kinematics of the northernmost section of the SVF.

3. **The third and last main point raised by R#2 is related to the discussion about the geodynamic engine of the SVFS linked to the indentation and CCW rotation of Adria.**

We note that we highlighted the connection of the complex kinematics of the SVFS with the geodynamic engine in different sections of the text and also for example in Figure 7.

In our interpretation, the SVFS is accommodating the indentation of the Adria plate. We did not consider the CCW rotation of Adria because it has no effects on the fault activity and kinematics (it explains the increasing shortening moving eastwards registered across the Southern Alps thrust fronts). However, we think that there is a little difference in respect to the interpretation given by the reviewer.

However, in one point our interpretation is different: in our model, the Posina triple junction is located inside the thrust belt and not at the most external front, so we think that the opening zipper model is a result of the presence of the back-stop of the Giudicarie belt that interacted with the northward propagation of the SVFS. In this sense, the right-lateral motion of the SVFS north of Posina should not be related to the differential propagation of the ESA thrust fronts.

**The main points raised by Reviewer #3 are the following:**

1. **The geometry of the indentation model that would be delimited by the Giudicarie system and not by the SVFS.**

We discussed more in detail this problem in the last paragraph of the Discussion chapter (#5). We also cited the papers suggested by the reviewer together with a recent paper just published by Solid Earth by Verwater et al. (2021). We suggest that there was an evolution of the indenter boundary through time that in the most recent phase would be delimited by the SVFS. The Adriatic plate indenter has been traditionally considered as delimited by the Giudicarie fault system to the west and the Pustertal-Gailtal fault to the north. Hence, the SVFS represented an intraplate structure crossing the Mesozoic Trento platform. However, we think that since the late Messinian the Adriatic block decoupled from its nearly stationary northwestern (Padanian) protuberance, and consequently the western indenter margin widened incorporating the SVFS, while leaving outside the contractional deformation the Lessini-Berici-Euganei foreland block.

2. **The timing of activity of the faults forming the zipper model that if not contemporaneous would indicate a truncation relationships of the conjugate faults and disregard the zipper structure.**

If the hypothesis of Reviewer #3 is true, all the SVF would be dextral (also south of Posina). However, we observe that the thrust front near Schio has a sinistral offset (see Figure 5), and also more to the south there are additional geological evidences of sinistral activity.

3. **The 30° transtensional analogue model performed by Fedorik et al. (2019).**

We described better in the text these experiments and their implications for our model.

4. **The GPS velocity field not showing deformation across the SVFS.**

Unfortunately, the geometry of the GPS network is not as dense as necessary to register the interseismic deformation connected to a locked fault. Besides, we expect the SVFS to move with low rates and the area deformed during the interseismic period to be very narrow across the fault itself. For these reasons we did not discuss this topic in the text.

**Point-by-point response to the reviews and list of all relevant changes made in the manuscript**

RC1

Page 2, Line 39: "… Carrillo et al., 2020), in the Northern-Central Apennines (Tavarnelli et al., 2001; Butler et al., 2006; Peacock et al., 2017), in the Southern Apennines foreland… ".

We followed the suggestion given by the reviewer and modified accordingly the text.

Page 5, Line 100: "… in the Veneto Plain (Fig. 1), itwas drawn…

OK

Page 7, Line 187: "… Southern Alps,  whose compressional deformation…".

OK

Page 10, Line 282: "… two of which  are presented in Fig. 2 (IDs 6 and 8)…"

OK

Page 10, Line 282 – and also Page 9, Fig. 2 – Fig. 2, IDs 6, referred to in the text, is not indicated in the Figure. The following is stated in the Figure caption: "The IDs 4 and 6 can be found in the sheets 36 Schio (Braga et al., 1968) and 49 Verona (Bosellini et al., 1968) of the 1: 100,000 scale Geological Map of Italy. See also Table 2", but this indication is not straightforwards for the reader (al least, for this reader). It would be useful if the Authors could attempt at providing, or sketching, or summarizing, the missing information within the manuscript. Should this not be possible, the Authors should clearly indicate the missing documentation directly in the text, as they have already done in the Figure caption.

We decided to add the reference to the web pages where the geological sheets are published, because it was not possible to publish them directly in the paper (or in the additional material).

Page 10, Line 303: "… are  S-directed blind thrusts…". The use of vergence is inappropriate here, since vergence is a property of folding, not faulting. The meaning of the sentence is still clear, but I would suggest to avoid the conceptually wrong term " S-vergent" and to replace it with the more correct "S-directed".

The term "..-verging" is often used when referring also to faults, however, we decided to follow the suggestion of the reviewer and changed the text.

Page 12, Line 351: "… from the SVFL, and  don't belong to it."

OK

Page 14, Line 379: "… with most of the events…".

OK

Page 16, Line 431: "… branchline…". In general this is spelt with two separate words in structural accounts:

branch line. I would suggest that this notation is used in the manuscript.

OK

Page 16, Line 438: "… Zampieri et al. (2003), in which work…". I would change this sentence as follows:

"… Zampieri et al. (2003), where the SVFS branch…".

OK

 "The study of… have been conducted by Fedorik et al. (2019) by means of…". I would rephrase this sentence as follows: "The study of… has been carried out by Fedorik et al. (2019) by means of…".

OK

Page 17, Line 476: "new thrust front of the Fig. 6c.".

OK

Page 18, Line 497: "… moderate to large earthquakes (e.g. Tavarnelli et al., 2001; Butler et al., 2006; Di Bucci et al., 2010).

OK

Page 19, Line 503. I would add another important cross-lineament in the northern-central Apennines, (as a separate case.d) whose history was, similarly, characterized by a strike-slip reversal reactivation, from sinistral to dextral: this reversal occurred along the Ancona-Anzio Line (or Olevano Antrodoco Fault), as described by Tavarnelli et al., 2001 and by Butler et al., 2006. Indeed, the following statement, already present in the text ("All these fault systems, during their long history, have seen different tectonic phases often characterised by opposite sense of shear (e.g. first right-lateral strike slip, then left-lateral strike-slip, or vice-versa"), perfectly apply to the history described by Tavarnelli et al. 2001 (see their Fig. 11a and 11b) and by Butler et al. 2006 (see their Fig. 10a and 10b).

We changed the text and added the new references, including other ones relevant to the Olevano-Antrodoco Line). We had to change also the following paragraph because introducing the Olevano-Antrodoco fault, these examples of inherited structures are not anymore taken only from the foreland areas. The Olevano-Antrodoco is in the Central Apennines thrust belt and can be considered as the lateral ramp of the Sibillini thrust developed along an inherited paleogeographic boundary. This is why at first we did not include it in our list.

Page 19, Line 515: "… accrud accommodated…

OK

**RC2**

Page 2, Line 56: please insert reference

We inserted some relevant references that mapped the SVFS also in the mountain area.

Page 3, Line 95: age of last activation?

This column of Table 1 shows the age of the different tectonic phases that characterized the SVFS and not the age of last activation.

Page 5, Line 100: Being the most prominent feature south of the Alps in the Veneto plain (Fig.1), the SVF was drawn.......

OK

Page 6, Line 164

OK

We changed this figure according to the comments of the reviewers.

Page 9, Figure 2 ID3: ID3: the figure it not clear. where are the displaced terraces? Do you know the age of these terraces? Holocene or Pleistocene?

We redrew the Figure ID3 using different symbols for the different order of terraces (according to the original figure of Pellegrini, 1988). The map shows that in the block west of the fault trace there are more terrace orders than in the block east of the fault. This is an indication of the differential uplift occurred between the two blocks and driven by the fault activity. In this sense, the terraces are not displaced, or at least there is no indication of displacement from the original paper, but they can be used as markers of the long-term throw of the fault that induced the differential uplift. Accordingly, we also added some explanations in the figure caption and corrected it and in the text.

Page 9, Line 265: I think that it would be better to include also the scraps/examples of the Schio and Verona geological sheets. May be in Supplementary material?

This could be a good idea, unfortunately it can be very hard to publish geological maps for copyright reasons, so we inserted in the caption, in the text and in the bibliography the reference to the public websites where the geological maps are published and freely accessible.

No, longitudinal to the river course, so crossing the fault trace from the western to the eastern block.

Page 10, Line 281: it would be interesting to see the examples.

One example is the ID 5 in Fig. 2. The other examples can be extrapolated from the geological maps.

OK

We changed the phrase according to the suggestion of the reviewer.

Page 11, Line 314: not clear.... tectonic activity in friuli and Veneto gave rise to widespread morphotectonic evidence also...see for example Galadini et al., 2005 or falcucci et al., 2018 where field geologic evidence is showed.

We modified the sentence inserting new references relative to morphotectonic, paleoseismological and geological evidence of the thrust activity.

We decided to keep the word elusive, because the faults sources of these earthquakes have not been identified yet.

We added the reference to the paper by Scardia et al. (2015), that was already in the citation list.

We changed this phrase according to the suggestion of the reviewer.

See our answer in this document.

Page 14, Figure 4: Please, if possible enlarge the figure.

We made all figures larger wherever possible

Page 14, Figure 4: TF and CF are left lateral strike slip fault while the n. 3 earthquake is dextral. Earthquake 7 and 9 are located in the left lateral sector of the SCF, but FMs are dextral.

See our answer to the main point #2 of RC2 in this document.

Page 15, Line 415: in the Italian-Slovenian Border region we observe transpressional and trantensional faults see for example Falcucci et al, 2018 , Bajc et al., 2001, Poli and Renner, 2004.

Yes, it's true, in that region there are the compressional structures belonging to the external Dinarides as well as the trascurrent faults of the Dinaric system. Both are active as a consequence of the convergence of Adria towards Europe (so they have the same geodynamic engine of the Eastern Southern Alps system).

In this sentence we wanted to point only to the complementary role played in our interpretation by the SVFS and the Dinaric system, so we did not give a full description of the tectonic framework of that area.

Page 15, Line 451: The Authors forget the geodynamic engine where SVFS develops. I recommend to highlight that the complex kinematics of the SVF is linked to indentation and counterclockwise rotation of Adria. As a consequence, in its southern part (foreland/plain) the SVF accompanies the indentation of Adria under the Alps and therefore acts as a left lateral strike slip fault, while in the northern part (up to the external front of the eastern Southern Alps) the SVF accompanies the southern propagation of the Southalpine chain, acting as a right lateral strike slip fault. Also these considerations are in agreement with Serpelloni et al., 2016

See our answer to the main point #3 of RC2 in this document.

Page 18, Figure 7: In contrast with the eastern portion of the eastern Southern Alps, where a set of right lateral strike slip faults (Dinarides) cuts and deforms the Southalpine front propagating within the Chain, the SVF passively borders the Southalpine western border acting as a right lateral trascurrent fault in the North and left lateral in the South.

We described the role played by the SVFS in the text.

**RC3**

Page 1, Line 31: This is not an English term and needs some explanation

The expression is taken from Platt & Passchier (2016) and Passchier & Platt (2017).

Page 2, Line 55: Lessini Mts., Berici Mts. is this abbreviation explained somewhere?

We changed Mts. (that is a common abbreviation for mountains) with the word Mountains.

Page 2, Line 55: There is a disparaity between the main segment of the SVFS, where preivious studies are cited, and decriptions are given, and the northern and southern segments, where neither references nor descriptions are given.

We added references to published papers dealing with the northern and southern segments.

Page 9, Line 91: I do not understand this sentence. Reformulate!

We reformulated the sentence.

Page 5, Line 105: Why "whole"? Most probably buried strands are found below foredeep deposits, and you refer to the southernmost part of the SVFS. Do you? Please clarify!

We changed the text trying to clarify the meaning of the sentence.

Page 5, Line 117: "Vicenza as a"? Complicated sentence, split into two."

We splitted the sentence in two.

Page 6, Line 148

OK

Page 6, Line 162: Unclear phrase. Reformulate! Or omit. You mean the Veneto plain?

We changed the text trying to clarify the sentence.

Page 8, Line 236: Which type of swarm? Earthquakes? Or a set auf faults? Clarify!

It is a set of faults, so we used "fault swarm".

Page 8, Line 242: I doubt that such a deep earthquake can be clearly connected to a surface fault havin only a few km of offset.

This is clearly a minor earthquake, that could have happened on a secondary fault related to the SVF. Nontheless, this is the largest event of the area and it was used as indicator of the kinematics of the fault. We preferred to cancel the hypocentral depth in this phrase since it is reported in Table 4).

Page 8, Line 252

We changed the sentence o clarify the concept.

Page 9, Figure 2: These are very nice kinematic indicators shown here. The red arrow points to a Riedel fault branching of a main fault, and there seem to be several on this surface. According to Petit (1987) such a geometry would conform with the RM-criterion, and in this case this fault would be dextral. Petit, J. P. (1987): Criteria for the sense of movement on fault surfaces in brittle rocks.- J. Struct. Geol., 9 (5-6): 597 - 608.

We agree that the fault plane bears several good kinematic indicators (macro and micro), as described in Fondriest et al. (2012). Undoubtedly, they point to a sinistral kinematics. The feature indicated by red arrow isn't a RM structure. The apparent similarity is due to the small dimension of the photo, which don't permit to resolve the structure.

Page 10, Line 281

OK

Page 10, Line 282: ID6 is a "Geological structure (drag fold)" according to Table 2 and not a seismic section. It is not shown in Fig. 2, therefore it is not "presented". ID8 are "Fractures of a travertine mound" and is not a seismic section either. However, it does highlight the extensional displacement. Is there a problem with the numbering? Reformulate and clarify!

We reformulated and corrected the numbering og the geologic evidence we referred to in the text.

 What does "local" refer to? Earthquakes at the SVFS? Or earthquakes in the respective towns?

With the term "local earthquake" we mean an earthquake with an epicenter (very probably) close to the town where it was felt. We also clarified this concept in the caption and in the text.

Page 16, Line 421: I am very surprised to hear about the presence of 'dolomitic marbles' in the Southern Alps in an area not adjacent to volcanites. I do not know the area, but probably this should be mentioned in the description further up, or omitted if it is not important.

We explained in the text why there are marbles in that area.

Page 16, Line 422: These corrugations are not "mega". This is the typical appearance of a corrugated fault surface.

OK

Page 17, Line 462-465: Unfortunately, Fig. 6 does not help to understand the model, and also the text is insufficient. This model needs more explanation.

We explained in the text the model.

Page 17, Line 468: I do not understand this phrase. Either kinematic modelling results in transtension, or not. What is "expected" here?

You are right. We deleted the word "expected".

Page 18, Figure 6: The 3D model does not help understanding geology, as this manuscript almost generally avoids geological descriptions. The reader has heard nothing about the the sedimentary succession of the Southern Alps, and therefore information about the Permian to Miocene horizons is not necessary.

This geological sketch is based on subsurface geological data and 3D reconstruction of fault geometry. We used it to compare with the model in Fig. 6c. The stratigraphic horizons of the 3D model were inserted as reference just to show the dip-slip component of the fault slip. Therefore, a description of the sedimentary succession is not necessary and would stretch the text too much.

Page 18, Line 483: See comment on last page. I do not understand, what a 30° transtensional model is. A fault can be transtensional, but it must be defined which fault.

We described the model in the text.

**Additional changes to the text**

- We added the reference to the paper by Masetti et al., 2012, that reports a seismic section crossing the SVFS. Consequently, we changed Table 1 and Table 2, and Figure 2 and Figure 4.
- We modified Figure 2 following the suggestions of Ref. #2 and #3. In particular we modified the panels ID1, ID2, ID3 and ID10.
- We modified Figure 7 adding an inset showing the geodynamic setting of the Adria indenter.
- Page 22, Line 564-577: we added a final sentence to the Discussion to describe in more detail the role played by the Adriatic indenter and the relative position of the SVFS within this framework. In this paragraph we added a reference to the recent paper by Verwater et al. 2021 (Solid Earth). We added this paper also to Table 1

- We added some reference to recent published papers (published after our first submission), like for example the paper by Atanackov et al., 2021.

---

## Author Response (AR2)

[revised manuscript text omitted]

**Commentato [PB1]:** We modified the Panels ID1 e ID2 inserting the traces of the striae; ID3 changing the symbols of the terrace orders; ID10 putting the TWT in the vertical scale; and the central panel adding a new trace of a seismic line published by Masetti et al., 2012.

[revised manuscript text omitted]